# BREAKING THE CORRELATION PLATEAU: ON THE OPTIMIZATION AND CAPACITY LIMITS OF ATTENTION-BASED REGRESSORS

**Jingquan Yan**\*, **Yuwei Miao**\*, **Peiran Yu, Junzhou Huang**
Computer Science and Engineering Department
University of Texas at Arlington
{jingquan.yan,yuwei.miao,peiran.yu,jzhuang}@uta.edu
\*Equal contribution.

## ABSTRACT

Attention-based regression models are often trained by jointly optimizing Mean Squared Error (MSE) loss and Pearson correlation coefficient (PCC) loss, emphasizing the magnitude of errors and the order or shape of targets, respectively. A common but poorly understood phenomenon during training is the *PCC plateau*: PCC stops improving early in training, even as MSE continues to decrease. We provide the first rigorous theoretical analysis of this behavior, revealing fundamental limitations in both optimization dynamics and model capacity. First, in regard to the flattened PCC curve, we uncover a critical conflict where lowering MSE (magnitude matching) can *paradoxically* suppress the PCC gradient (shape matching). This issue is exacerbated by the softmax attention mechanism, particularly when the data to be aggregated is highly homogeneous. Second, we identify a limitation in the model capacity: we derived a PCC improvement limit for *any* convex aggregator (including the softmax attention), showing that the convex hull of the inputs strictly bounds the achievable PCC gain. We demonstrate that data homogeneity intensifies both limitations. Motivated by these insights, we propose the Extrapolative Correlation Attention (ECA), which incorporates novel, theoretically-motivated mechanisms to improve the PCC optimization and extrapolate beyond the convex hull. Across diverse benchmarks, including challenging homogeneous data setting, ECA consistently breaks the PCC plateau, achieving significant improvements in correlation without compromising MSE performance.

## 1 INTRODUCTION

The attention mechanism is a powerful tool for regression tasks where each input sample comprises multiple elements, such as tokens or image patches, and the elements' embeddings are aggregated using attention mechanism to predict a continuous target (Lee et al., 2019; Martins et al., 2020; Born & Manica, 2023). This paradigm is widely used in areas like digital pathology, time-series prediction and emotional analysis (Jiang et al., 2023; Ni et al., 2023; Zhang et al., 2023). For example, in video-based sentiment analysis, the attention mechanism is used to aggregate frame embeddings within a video clip to predict emotion variables, such as sentiment intensity, arousal, and valence (Truong & Lauw, 2019; Xie et al., 2024).

Two characteristics are common in such applications. First, the in-sample data frequently exhibits **higher homogeneity** than cross-sample data. For example, in pathology images, nearby regions tend to share more similar tissue or cell types than distant regions. Likewise, in video-based sentiment analysis, frames from the same clip are more similar than frames from different clips. Second, the regression performance is not simply evaluated by **magnitude** (e.g., Mean Squared Error (MSE)) but also by **shape**—the relative ordering of predictions, measured by the Pearson Correlation Coefficient (PCC). In many settings, capturing the correct correlations matters more than predicting the exact values (Pandit & Schuller, 2019). For instance, in spatial transcriptomics, capturing the relative trend of gene expression (*e.g.*, co-expression) is more informative than predicting exact magnitudes (Xiao et al., 2024). To emphasize shape while retaining magnitude performance, models are commonly

Figure 1: (a) Illustration of a video-based sentiment analysis example. A sample is considered homogeneous when its within-sample dispersion $\tilde{\sigma}$ is below $\sigma_0$. (b) A convex attention yields an aggregated embedding inside the convex hull of the sample's embeddings. (C) Our ECA extrapolates beyond the hull to amplify within-sample contrasts.

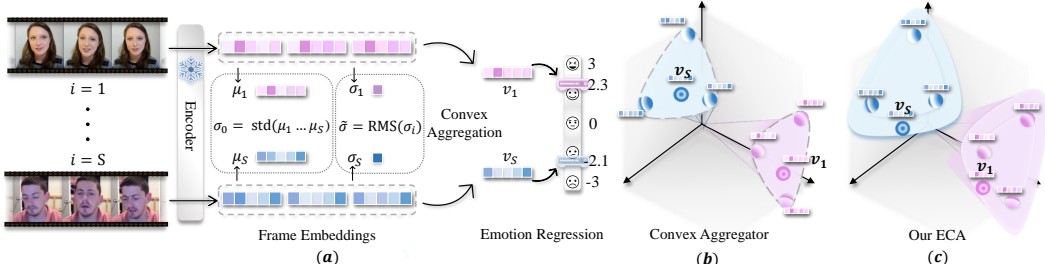

Frame Embeddings
(a)        Emotion Regression    Convex Aggregator    Our ECA
(b)              (c)

trained with a joint loss function, such as $\mathcal{L}_{\text{total}} = \mathcal{L}_{\text{MSE}} + \lambda_{\text{PCC}}(1 - \rho)$ where $\rho$ is the PCC and $\lambda_{\text{PCC}}$ is a hyperparameter weight (Liu et al., 2022; Zhang et al., 2023; Zhu et al., 2025).

However, such joint training strategy applied to attention-based regression model frequently exhibits a puzzling *PCC plateau*, even with large PCC weight $\lambda_{\text{PCC}}$. As shown in Figure 2, the PCC curve's slope flattens early in training, failing to improve further even while the MSE continues to decrease. This empirical phenomenon is particularly severe with high in-sample homogeneity data. The underlying mechanisms driving this plateau remain unclear and it raises a critical question: why does the regression model fail to optimize the correlation effectively?

In this work, we provide the first theoretical investigation into this question, analyzing the limitations of standard attention from two distinct perspectives: **optimization dynamics** and **model capacity**.

**Limitation 1: Conflict in Optimization Dynamics.** Our analysis reveals a direct relation between the decrease of MSE and the flattening of the PCC curve in the attention-based regression model. Using a decomposition of MSE (Proposition 2.1), we show the following. Although training continues to reduce MSE by matching the predictions' mean and standard deviation, the gradient of the correlation term is paradoxically attenuated, flattening the PCC curve. This effect is amplified by *in-sample homogeneity*: when embeddings within a sample are similar, the within-sample dispersion term in the PCC gradient shrinks and further suppresses the magnitude of the gradient.

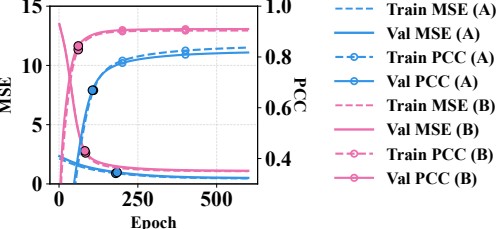

Figure 2: PCC plateau example: PCC flattens at early epoch while MSE continues to decrease. The plateau becomes more noticeable as within-sample homogeneity increases, specifically, dataset A is more homogeneous than B.

**Limitation 2: Model Capacity of Convex Aggregation.** Notice that softmax attention is a convex combination of element embeddings; we compare convex attention aggregation to mean-pooling aggregation and derive an upper bound on the achievable PCC gain (Theorem 2.2) depending on the radius of convex hull formed by the in-sample embeddings. Greater in-sample homogeneity will shrink the convex hull and tightens this bound, yielding a fundamental capacity limit for PCC gain.

**Extrapolative Correlation Attention.** Motivated by these theoretical insights, we propose Extrapolative Correlation Attention (ECA), a novel framework designed to overcome these limitations. To mitigate the optimization conflict in Limitation 1, ECA introduces a Dispersion-Normalized PCC loss to counteracts the attenuation effect, restoring the magnitude of correlation gradient. We also employ Dispersion-Aware Temperature Softmax to prevent the gradient collapse and near-uniform attention distribution under in-sample homogeneity. To address Limitation 2, ECA introduces Scaled Residual Aggregation, which allows the model to extrapolate beyond the convex hull, mitigating the PCC gain limitation induced by convex aggregation mechanism.

**Contributions.** We theoretically investigate the PCC plateau when training attention-based regression model with a joint MSE+PCC loss. We identify two limitations in optimization dynamics and in model capacity.

- **Optimization Dynamics.** We identify a conflict in optimization dynamics: as MSE optimization increases $\sigma_{\hat{y}}$ to match $\sigma_y$, the magnitude of the PCC loss gradient scales with $1/\sigma_{\hat{y}}$ leading to a flattened PCC curve during training. In addition, We show that within-sample homogeneity further shrinks the PCC gradient through the Jacobian of softmax attention. Both parts contribute to the PCC plateau phenomenon.
- **Model Capacity.** We prove that for any convex aggregator (*e.g.*, softmax attention), the achievable PCC improvement over mean-pooling can be bounded by the within-sample data homogeneity.

To address these limitations, we introduce ECA, a novel attention-based regression framework with three components:

- Motivated by the optimization limitation, we re-scale the PCC objective to counteract the $1/\sigma_{\hat{y}}$ factor, restoring correlation gradients magnitude while minimizing MSE.
- Motivated by the convex aggregator limitation, we incorporate a Scaled Residual Aggregation that enables controlled extrapolation beyond the convex hull.
- To handle within-sample homogeneity, we propose a Dispersion-Aware Temperature Softmax to prevent attention collapse and amplify the informative contrasts within homogeneous samples.

Across diverse regression benchmarks, ECA consistently overcomes the PCC plateau and achieves higher correlation gains while maintaining competitive MSE. Ablation studies confirm the contribution of each component.

## 2 THEORETICAL ANALYSIS OF CORRELATION LEARNING IN ATTENTION-BASED AGGREGATION

We theoretically analyze why the PCC plateau occurs when training attention-based regression with a joint MSE+PCC loss. First, we relate MSE and PCC through a decomposition that motivates joint training. Then we prove that softmax attention, as a convex aggregator, limits both the optimization dynamics and the model's expressive capacity. These limits hinder correlation learning and explain the observed PCC plateau.

### 2.1 PROBLEM SETUP

**Data Notation.** We consider a regression dataset with a batch of $S$ samples $(x_s, y_s)$. Each sample $x_s$ is a set of $n_s$ element with embeddings $\mathbf{h}_s = \{\mathbf{h}_{si}\}_{i=1}^{n_s}$, where $\mathbf{h}_{si} \in \mathbb{R}^d$. Let $y_s, \hat{y}_s \in \mathbb{R}$ be the ground-truth target and model prediction for $s_{\text{th}}$ sample. The batch-level empirical means are $\mu_y, \mu_{\hat{y}}$ and standard deviations are $\sigma_y, \sigma_{\hat{y}}$. Define centered targets and predictions as $a_s := y_s - \mu_y$ and $b_s := \hat{y}_s - \mu_{\hat{y}}$.

**Attention-based Aggregation.** The attention-based model processes each input sample to produce a scalar prediction. An attention scoring function $f_{\text{attn}}(\cdot)$ (*e.g.*, KQV dot-product similarity or gating mechanism) scores each embedding in $\mathbf{h}_s$ and produces attention logits $\mathbf{z}_s = f_{\text{attn}}(\{\mathbf{h}_{si}\}_{i=1}^{n_s}) \in \mathbb{R}^{n_s}$ with entries $z_{si} = [\mathbf{z}_s]_i \in \mathbb{R}$. Softmax converts these logits to positive attention weights $\boldsymbol{\alpha}_s = \text{Softmax}(\mathbf{z}_s)$ on the probability simplex with entries $\alpha_{si} = [\boldsymbol{\alpha}_s]_i \in \mathbb{R}$ and $\sum_{i=1}^{n_s} \alpha_{si} = 1$. The sample-level embedding is the convex combination $\mathbf{v}_s = \sum_{i=1}^{n_s} \alpha_{si} \mathbf{h}_{si} \in \mathbb{R}^d$. Finally, a linear regression head with weights $\mathbf{w} \in \mathbb{R}^d$ and bias $c \in \mathbb{R}$ produces the scalar prediction: $\hat{y}_s = \mathbf{w}^\top \mathbf{v}_s + c$. Note that this formulation is backbone-agnostic: $\{\mathbf{h}_{si}\}$ represents the features at the final layer of any deep architecture (*e.g.*, a multi-layer Transformer). Since these models typically derive the final prediction with a convex attention pooling (*e.g.*, [CLS] token aggregation) followed by a linear projection, our analysis of the attention-based aggregation and its interaction with joint MSE and PCC optimization applies regardless of the depth or complexity of the preceding backbone. We provide a detailed discussion on the applicability to deep architectures in Appendix E.

**Learning Objective.** We measure Pearson correlation between targets and predictions over the batch by $\rho := \frac{\text{Cov}(\mathbf{y}, \hat{\mathbf{y}})}{\sigma_{\mathbf{y}} \sigma_{\hat{\mathbf{y}}}}$. To optimize both magnitude and correlation of the prediction, we use the popular joint loss $\mathcal{L}_{\text{total}} := \text{MSE}(\mathbf{y}, \hat{\mathbf{y}}) + \lambda_{\text{PCC}}(1 - \rho)$ where $\lambda_{\text{PCC}} \geq 0$ balances the two terms.

**Homogeneity Measures.** For each sample, let the in-sample mean be $\boldsymbol{\mu}_s = \frac{1}{n_s} \sum_{j=1}^{n_s} \mathbf{h}_{sj}$. We quantify within-sample homogeneity by (i) the in-sample dispersion $\sigma_s = \sqrt{\frac{1}{n_s} \sum_{j=1}^{n_s} \|\mathbf{h}_{sj} - \boldsymbol{\mu}_s\|_2^2}$, and (ii) the convex-hull radius $R_s := \max_i \|\mathbf{h}_{si} - \boldsymbol{\mu}_s\|_2$. These quantities are related by inequality $\sigma_s \leq R_s \leq \sqrt{n_s}\, \sigma_s$ with proof in Lemma D.3. Cross-sample homogeneity is summarized by the root-mean-square (RMS) values $\tilde{\sigma} := \left(\frac{1}{S} \sum_{s=1}^{S} \sigma_s^2\right)^{1/2}$ and $\tilde{R} := \left(\frac{1}{S} \sum_{s=1}^{S} R_s^2\right)^{1/2}$.

## 2.2 PRELIMINARIES: THE INTERPLAY BETWEEN MSE AND PCC

**Proposition 2.1** (MSE Mean–std–correlation Decomposition). *The MSE between $y$ and $\hat{y}$ can be decomposed as:*

$$\mathrm{MSE}(\mathbf{y}, \hat{\mathbf{y}}) = \underbrace{(\mu_{\hat{\mathbf{y}}} - \mu_{\mathbf{y}})^2}_{\text{mean matching}} + \underbrace{(\sigma_{\hat{\mathbf{y}}} - \sigma_{\mathbf{y}})^2}_{\text{std matching}} + \underbrace{2\,\sigma_{\mathbf{y}}\,\sigma_{\hat{\mathbf{y}}}\,(1 - \rho)}_{\text{weighted correlation}}. \tag{1}$$

**Lemma 2.1** (Scaling-invariance of PCC). *Let $m \geq 0$ and $n \in \mathbb{R}$. $\mathrm{PCC}(y, m\hat{y} + n) = \mathrm{PCC}(y, \hat{y})$.*

We defer the proof for Proposition 2.1 and Lemma 2.1 to Appendix A.

**Remark 2.1.** *Proposition 2.1 and lemma 2.1 are model-independent and hold for any regressor.*

**Remark 2.2** (The PCC Plateau). *By Proposition 2.1, minimizing MSE jointly targets (i) mean matching, (ii) standard deviation matching, and (iii) weighted correlation. Because MSE is sensitive to affine transformations while PCC is invariant (Lemma 2.1), optimization can reduce MSE by mainly adjusting mean and scale, with little improvement on correlation. This explains why MSE may keep decreasing while PCC plateaus, and motivates adding the explicit correlation term $\lambda_{PCC}(1 - \rho)$ to the objective.*

The discussion above provides intuitions for the PCC plateau phenomenon. We further empirically validate this phenomenon on 8 UCI regression datasets using multi-layer Transformers (Appendix F). Figure 7 shows a consistent pattern across all tasks: PCC curve flattens before MSE convergence. This consistent pattern confirms that "PCC plateau" is a general correlation learning failure mode under attention-based joint optimization setting. We now provide a formal analysis by characterizing the PCC gradient in attention-based regression models.

## 2.3 OPTIMIZATION DYNAMICS: THE CORRELATION GRADIENT BOTTLENECK

We study how the gradients propagate through the attention aggregator when optimizing the joint loss $\mathcal{L}_{\text{total}} = \mathcal{L}_{\text{MSE}} + \lambda_{\text{PCC}}(1 - \rho)$. We derive the gradients of both MSE and PCC with respect to the attention logits $z_{si}$ to understand the optimization dynamics.

**Lemma 2.2** (Softmax Aggregator Jacobian). *The derivative of the aggregated embedding $\mathbf{v}_s$ with respect to a pre-softmax logit $z_{si}$ is $\partial \mathbf{v}_s / \partial z_{si} = \alpha_{si}(\mathbf{h}_{si} - \mathbf{v}_s)$ and consequently, $\partial \hat{y}_s / \partial z_{si} = \alpha_{si}\mathbf{w}^\top(\mathbf{h}_{si} - \mathbf{v}_s)$.*

**Theorem 2.1** (Gradient of PCC w.r.t. Attention Logits). *The derivative of Pearson correlation $\rho$ with respect to a pre-softmax logit $z_{si}$ is*

$$\frac{\partial \rho}{\partial z_{si}} = \frac{1}{S\sigma_{\hat{y}}} \left(\frac{a_s}{\sigma_y} - \rho\frac{b_s}{\sigma_{\hat{y}}}\right) \alpha_{si}\mathbf{w}^\top(\mathbf{h}_{si} - \mathbf{v}_s). \tag{2}$$

To understand the interplay during joint optimization, we also examine the gradient of the MSE loss.

**Lemma 2.3** (Gradient of MSE w.r.t. Attention Logits). *The derivative of MSE with respect to a pre-softmax logit $z_{si}$ is*

$$\frac{\partial \mathcal{L}_{\text{MSE}}}{\partial z_{si}} = \frac{2}{S}(\hat{y}_s - y_s)\,\alpha_{si}\mathbf{w}^\top(\mathbf{h}_{si} - \mathbf{v}_s). \tag{3}$$

**Gradient Decomposition and the Optimization Conflict.** Comparing the PCC gradient (Equation (2)) and the MSE gradient (Equation (3)), we observe they share the same *local* structure factor $L_{si} := \alpha_{si}\,\mathbf{w}^\top(\mathbf{h}_{si} - \mathbf{v}_s)$, which governs attention adjustment within sample $s$. The difference lies entirely in the *global* scaling factors which depend on overall batch statistics:

$$\frac{\partial \mathcal{L}_{\text{MSE}}}{\partial z_{si}} = \frac{1}{S} g_s^{\text{MSE}} L_{si}, \quad \text{where } g_s^{\text{MSE}} := 2(\hat{y}_s - y_s), \tag{4}$$

$$\frac{\partial \rho}{\partial z_{si}} = \frac{1}{S} g_s^{\text{PCC}} L_{si}, \quad \text{where } g_s^{\text{PCC}} := \frac{1}{\sigma_{\hat{y}}} \left( \frac{a_s}{\sigma_y} - \rho \frac{b_s}{\sigma_{\hat{y}}} \right). \tag{5}$$

The relative impact of MSE versus PCC optimization is determined by the ratio of these global factors. We analyze this ratio using their Root Mean Square (RMS) values across the batch.

**Corollary 2.1** (PCC/MSE Gradient Ratio Decay). *Assuming $\rho \in [0, 1]$, the RMS ratio of the global scaling factors across the batch is bounded by:*

$$r_{\text{global}} := \frac{\text{RMS}_s(g_s^{\text{PCC}})}{\text{RMS}_s(g_s^{\text{MSE}})} \leq \frac{1}{2\sqrt{\sigma_y}} \cdot \frac{1}{\sigma_{\hat{y}}^{3/2}}. \tag{6}$$

Corollary 2.1 identifies a gradient bottleneck where the PCC signal attenuates relative to MSE at a rate of $\mathcal{O}(1/\sigma_{\hat{y}}^{3/2})$. This bound is empirically validated in Figure 3 on synthetic dataset , illustrating how magnitude matching dominates the optimization dynamics. We further analyze the magnitude of the PCC gradient alone, which reveals dependence on in-sample homogeneity.

**Corollary 2.2** (PCC Gradient Magnitude Bound). *The magnitude of the PCC gradient in Theorem 2.1 can be bounded by*

$$\left| \frac{\partial \rho}{\partial z_{si}} \right| \leq \underbrace{\frac{1}{\sigma_{\hat{y}}}}_{\substack{\text{prediction} \\ \text{deviation}}} \cdot \underbrace{\frac{4\sqrt{n_s(S-1)}}{S}}_{\text{batch scale}} \cdot \underbrace{\|\mathbf{w}\|_2}_{\substack{\text{regression} \\ \text{weights}}} \cdot \underbrace{\sigma_s}_{\substack{\text{in-sample} \\ \text{dispersion}}}. \tag{7}$$

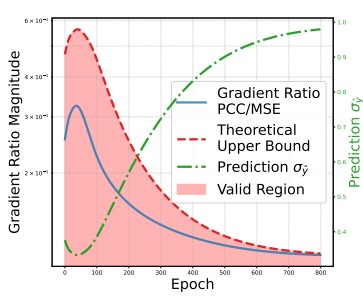

Detailed derivations for Lemma 2.2, Theorem 2.1 are provided in Appendix B.1. Proofs for Lemma 2.3, Corollary 2.1, and Corollary 2.2 are provided in Appendix C.

**Remark 2.3** (The Two Bottlenecks of Softmax Attention for Correlation). *The gradient analysis reveals two key bottlenecks in optimization dynamics that drive PCC plateaus:*

**1. Dominance of the MSE Gradient.** *Corollary 2.1 reveals a critical conflict in the joint optimization: the ratio of the PCC gradient magnitude relative to the MSE gradient magnitude decays rapidly at a rate of $\mathcal{O}(1/\sigma_{\hat{y}}^{3/2})$. Training with the joint loss minimizes the MSE std-matching term in Equation (1), which drives $\sigma_{\hat{y}}$ toward the target standard deviation $\sigma_y$. As $\sigma_{\hat{y}}$ typically increases during early training (see Figure 4), the relative contribution of the PCC gradient diminishes significantly. Consequently, the optimization becomes dominated by the MSE objective (magnitude matching), effectively downplaying PCC optimization (shape matching) and causing the plateau, even when the PCC loss weight $\lambda_{PCC}$ is large. This motivates optimization strategies that counteract this rapid attenuation.*

**2. Dependence on Within-sample Homogeneity.** *The gradient bound in Corollary 2.2 is proportional to the in-sample dispersion $\sigma_s$. When a sample's elements are homogeneous, $\sigma_s$ is small and the PCC gradient magnitude reduces, effectively hindering improvements to PCC via attention adjustment. Furthermore, since attention scoring functions are generally continuous, homogeneous inputs lead to low-variance logits $z_{si}$. Under fixed-temperature softmax, this results in near-uniform weights $\alpha_{si} \approx 1/n_s$, suppressing per-sample selectivity. These effects motivate mechanisms that adapt attention sensitivity to the in-sample dispersion.*

Figure 3: Validation of gradient ratio (PCC/MSE) decay. The RMS ratio of PCC vs. MSE gradients (blue) is strictly constrained by the theoretical upper bound (red). The increase in prediction dispersion $\sigma_{\hat{y}}$ (green) during training drives the attenuation of the PCC gradient signal relative to the MSE gradient.

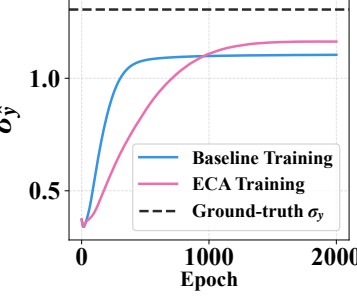

Figure 4: The standard deviation of predictions $\sigma_{\hat{y}}$ increases during training to match the standard deviation of labels $\sigma_y$ under MSE loss.

## 2.4 MODEL CAPACITY: THE PCC CEILING OF CONVEX AGGREGATION

Beyond optimization dynamics, we study the limits of the aggregator's expressivity. Softmax attention performs convex combinations, restricting aggregated embedding $\mathbf{v}_s$ to the convex hull of the in-sample embeddings $\{\mathbf{h}_{si}\}$. We study how much PCC improvement can be achieved by any convex aggregator over mean-pooling and show a capacity limit governed by in-sample homogeneity.

**Prediction Decompositon.** We decompose the sample embedding $\mathbf{v}_s$ relative to the mean-pooling embedding $\boldsymbol{\mu}_s$. The prediction can be decomposed as: $\hat{y}_s = \underbrace{(\mathbf{w}^\top \boldsymbol{\mu}_s + c)}_{\bar{y}_s} + \underbrace{\mathbf{w}^\top (\mathbf{v}_s - \boldsymbol{\mu}_s)}_{\Delta \hat{y}_s}$, where

$\bar{y}_s$ is the prediction using mean-pooling aggregation, and $\Delta \hat{y}_s$ is the attention-induced perturbation.

**Theorem 2.2** (PCC Gain Bound for Convex Attention). *Let $\sigma_0 := \mathrm{std}_s(\bar{y}_s)$ be the standard deviation of the mean-pooling predictions; define baseline $\rho_0 := \mathrm{PCC}(\mathbf{y}, \bar{\mathbf{y}})$ and $\rho$ is the PCC after applying any convex attention mechanism. Provided $\|\mathbf{w}\|_2 > 0$ and $\tilde{R} < \sigma_0 / \|\mathbf{w}\|_2$,*

$$|\rho - \rho_0| \leq \frac{2\tilde{R}}{\sigma_0 / \|\mathbf{w}\|_2 - \tilde{R}}. \tag{8}$$

Detailed proof is provided in Appendix D.

**Remark 2.4.** *Theorem 2.2 shows that the PCC improvement bound of any convex attention depends only on the ratio between convex hull radius $\tilde{R}$ (reflecting in-sample homogeneity) and the normalized standard deviation of mean-pooling baseline ($\sigma_0 / \|\mathbf{w}\|_2$). The bound is scale-invariant and independent of the regression head magnitude since $\sigma_0 / \|\mathbf{w}\|_2 = \mathrm{std}_s(\mathbf{w}^\top \boldsymbol{\mu}_s) / \|\mathbf{w}\|_2 = \mathrm{std}_s\big((\mathbf{w}^\top / \|\mathbf{w}\|_2)\boldsymbol{\mu}_s\big)$. When in-sample homogeneity is high ($\tilde{R}$ is small), no convex attention can substantially increase correlation. The limitation arises because $\mathbf{v}_s$ is confined to the convex hull; a small hull restricts the magnitude of adjustments to the aggregated embedding, thereby limiting the potential PCC improvement and motivating mechanisms that can extrapolate beyond it.*

## 2.5 SUMMARY OF THEORETICAL INSIGHTS

Our analysis reveals that the difficulty of optimizing PCC using softmax attention stems from two aspects: optimization dynamics and model capacity. Remark 2.3 highlights the vanishing PCC gradients due to cross-sample dispersion attenuation $1/\sigma_{\hat{y}}$ and weak in-sample dispersion. Remark 2.4 further shows that any convex aggregator is restricted to the convex hull, which limits the possible PCC gain by ratio $\tilde{R}/(\sigma_0 / \|\mathbf{w}\|_2)$. Together, these effects explain the plateau and motivate a novel attention mechanism proposed in the next section that addresses the identified bottlenecks accordingly.

## 3 EXTRAPOLATIVE CORRELATION ATTENTION (ECA)

Our analysis has identified fundamental limitations in softmax attention for optimizing joint MSE+PCC objective. To address these issues, we propose Extrapolative Correlation Attention (ECA), a novel drop-in attention module for regression that enhances both optimization and expressivity. ECA incorporates three components: (i) Scaled Residual Aggregation to break the convex hull constraint; (ii) Dispersion-Aware Temperature Softmax, to avoid gradient collapse; and (iii) Dispersion-Normalized PCC Loss, which compensates the $1/\sigma_{\hat{y}}$ attenuation in correlation gradients.

### 3.1 BREAKING THE CONVEX HULL WITH SCALED RESIDUAL AGGREGATION (SRA)

Theorem 2.2 shows that any convex attention mechanism is capacity-limited in correlation improvement because the aggregated embedding $\mathbf{v}_s$ lies inside the convex hull of the in-sample embeddings $\{\mathbf{h}_{si}\}$. This PCC gain bound is especially tighter when in-sample dispersion is low. To relax this limit, we introduce Scaled Residual Aggregation (SRA): instead of a strict convex aggregation, the model extrapolates along the residual $(\mathbf{h}_{si} - \boldsymbol{\mu}_s)$, allowing $\mathbf{v}_s$ to move beyond the convex hull.

Given the mean embedding $\boldsymbol{\mu}_s = 1/n_s \sum_i \mathbf{h}_{si}$, we define the residual $\Delta \mathbf{v}_s$ as the attention-weighted deviation from the mean: $\Delta \mathbf{v}_s := \sum_i \alpha_{si}(\mathbf{h}_{si} - \boldsymbol{\mu}_s)$. SRA scales this residual by a

learnable factor $\gamma_s \geq 1$:

$$\mathbf{v}_s^{\text{ECA}} = \boldsymbol{\mu}_s + \gamma_s \cdot \Delta\mathbf{v}_s = \boldsymbol{\mu}_s + \gamma_s \sum_i \alpha_{si}(\mathbf{h}_{si} - \boldsymbol{\mu}_s). \tag{9}$$

We parameterize $\gamma_s$ using a small, sample-specific MLP conditioned on the mean embedding $\boldsymbol{\mu}_s$ and use a shifted Softplus activation to ensure $\gamma_s \geq 1$, that is $\gamma_s = 1 + \text{Softplus}(\text{MLP}_{\theta_\gamma}(\boldsymbol{\mu}_s))$.

The factor $\gamma_s$ allows the model to amplify weak in-sample contrasts. When $\gamma_s = 1$, SRA reduces to standard convex attention aggregation. For $\gamma_s > 1$, the model extrapolates beyond the convex hull, and more importantly, it breaks the convexity constraint. In standard attention, the deviation $\|\Delta\mathbf{v}_s\|$ is bounded by the radius of the convex hull $R_s$. SRA expands the reachable space by increasing the effective radius and fundamentally bypasses the capacity limit derived for convex aggregators in Theorem 2.2. In practice we optionally clip $\gamma_s$ at a maximum (e.g., $\gamma_{\max} = 2$) or add a regularizer $\mathcal{L}_\gamma = \frac{\lambda_\gamma}{S} \sum_s (\gamma_s - 1)^2$ to discourage excessive scaling.

### 3.2 Dispersion-Aware Temperature Softmax (DATS)

While SRA enables extrapolation beyond the convex hull, the model still needs a informative direction to extrapolate. In homogeneous samples, standard softmax produces flat attention $\alpha_{si} \approx 1/n_s$ (Remark 2.3), which pulls the aggregated embedding toward $\boldsymbol{\mu}_s$ and makes the residual in Equation (9) small ($\Delta\mathbf{v}_s \approx 0$). With small residual, SRA provides little benefit. To address this, we introduce Dispersion-Aware Temperature Softmax (DATS), which adapts the attention temperature to the in-sample dispersion, sharpening attention when homogeneity is high:

We modify the softmax attention for Equation (9) with a sample-specific temperature $\tau_s$ reflecting within-sample dispersion:

$$\alpha_{si} = \text{softmax}\left(\frac{z_{si}}{\tau_s}\right), \quad \tau_s = T_{\min} + \beta\sqrt{\frac{1}{n_s}\sum_{1 \leq i \leq n_s}\|\mathbf{h}_{si} - \boldsymbol{\mu}_s\|^2}. \tag{10}$$

Here $T_{\min} > 0$ lower-bounds the temperature for stability, and $\beta \geq 0$ is a hyperparameter that controls sensitivity. When embeddings within a sample are homogeneous, $\tau_s$ has lower value so small differences between logits $z_{si}$ become sharper after attention, yielding a meaningful deviation $\Delta\mathbf{v}_s$ that SRA can effectively amplify.

### 3.3 Stabilizing Optimization: Dispersion-Normalized PCC Loss (DNPL)

Theorem 2.1 shows that the correlation gradient is attenuated by $1/\sigma_{\hat{y}}$. As MSE optimization improves standard deviation matching, $\sigma_{\hat{y}}$ increases, which shrinks the PCC gradient and contributes to a PCC plateau. We counteract this attenuation effect with a Dispersion-Normalized PCC Loss (DNPL), which rescales the PCC term by the current prediction standard deviation while blocking its gradient:

$$\tilde{\mathcal{L}}_{\text{PCC}} = \text{StopGrad}(\sigma_{\hat{y}}) \cdot (1 - \rho). \tag{11}$$

The $\text{StopGrad}(\cdot)$ operation ensures we only adjust the gradient magnitude to counteract the attenuation, while leaving the learning objective's stationary points unchanged.

### 3.4 Overall Objective

The complete ECA framework, including SRA and DATS, is fully differentiable and can be trained end-to-end. The overall learning objective combines the primary regression loss (MSE), the normalized PCC loss (DNPL), and the extrapolation regularizer:

$$\mathcal{L}_{\text{Total}} = \mathcal{L}_{\text{MSE}} + \lambda_{\text{PCC}} \cdot \tilde{\mathcal{L}}_{\text{PCC}} + \mathcal{L}_\gamma. \tag{12}$$

## 4 Related Works

**Correlation Learning and Optimization.** Correlation is a core metric in biological and medical areas (Langfelder & Horvath, 2008; Lawrence & Lin, 1989). In these domains, PCC is a standard

criterion for evaluating regression performance (Kudrat et al., 2025; Long et al., 2023). Because PCC is differentiable, it is often optimized directly as a loss in regression pipelines (Kudrat et al., 2025; Avants et al., 2008). From a multi-task perspective, many works combine PCC with MSE to balance the prediction magnitude and shape (Yang et al., 2023; Liu et al., 2022; Balakrishnan et al., 2019). However, the interaction dynamics between MSE and PCC under joint optimization remain underexplored, motivating our analysis of gradient coupling and the observed PCC plateau.

**Softmax Attention Aggregation.** Softmax attention is a cornerstone component in the backbones of many representative regression models (Zhou et al., 2021; Gorishniy et al., 2021; Kim et al., 2019). Despite its empirical success, recent theory has revealed expressivity limits of softmax mappings in related contexts (Yang et al., 2017; Kanai et al., 2018; Bhojanapalli et al., 2020). However, to our knowledge, no prior work analyzes the model capacity of softmax attention in terms of upper bounds on achievable PCC improvements, especially with high in-sample homogeneity data.

# 5 EXPERIMENTS

We evaluate our ECA on four settings, including the challenging high in-sample homogeneity tasks. The evaluations consist of: (i) a synthetic regression dataset with controllable in-sample homogeneity; (ii) three representative tabular regression benchmarks from the UCI ML Repository (Asuncion et al., 2007); (iii) a clinical pathology dataset for spatial transcriptomic prediction, where nearby regions exhibit *high homogeneity*; and (iv) a multimodal sentiment analysis (MSA) dataset, where consecutive video frames are *highly homogeneous*.

As ECA is a drop-in replacement for softmax attention, we integrate it into existing attention-based regression models for each benchmark to measure the performance improvement. More details are provided in Appendix G.

## 5.1 EXPERIMENTAL SETUP

**Synthetic Dataset.** We construct a synthetic dataset to validate our theory and proposed ECA method. We synthetic $N$ samples, each with $K$ element embeddings in $D$ dimensions as input samples. In each sample, one key element carries signal along a fixed unit direction $\mathbf{w}^*$, while the remaining $K-1$ background elements cluster around a shared sample mean. The label $\mathbf{y}$ is the projection of the sample mean onto $\mathbf{w}^*$ with a term proportional to the key strength and small additive noise. We control within-sample homogeneity via $\eta$ (larger $\eta$ means the key deviates further from the mean), yielding four homogeneity levels $\tilde{\sigma} \in \{0.10, 0.24, 0.42, 0.73\}$ where lower $\tilde{\sigma}$ indicates higher homogeneity. We compare regression model with one layer of ECA to one layer of standard softmax attention and report MSE and PCC.

**UCI ML Repository Datasets.** We evaluate on three representative tabular regression benchmarks: Appliance Energy Prediction (28 features, 1 target) (Candanedo, 2017), Online News Popularity (58 features, 1 target) (Fernandes & Sernadela, 2015), and Superconductivity (81 features, 1 target) (Hamidieh, 2018). We integrate ECA into the attention layer of the FT-Transformer (Gorishniy et al., 2021) and report mean absolute error (MAE), MSE, and PCC.

**Spatial Transcriptomic Dataset.** We test spatial transcriptomics prediction from pathology images on the 10xProteomic dataset (10x Genomics, 2025; Yang et al., 2023), which contains $32,032$ slide-image patches paired with gene-expression measurements of breast-cancer slides. We follow the data processing and experimental settings of the EGN baseline (Yang et al., 2023), which jointly optimize MSE+PCC loss. We adopt ECA methods onto the EGN baseline and report MSE, PCC@F, PCC@S, and PCC@M as evaluation metrics. The training set exhibits high in-sample embedding homogeneity with $\tilde{\sigma} = 0.068$ versus cross-sample $\sigma_0 = 0.164$.

**Multimodal Sentiment Analysis (MSA) Dataset.** We use MOSI (Zadeh et al., 2016), a standard MSA benchmark consists of $2,199$ monologue video clips with audio and visual inputs. As illustrated in Figure 1, consecutive frames within a clip are more similar than frames across clips. Quantitatively, the video frames shows strong within-sample homogeneity with $\tilde{\sigma} = 0.098$ versus cross-sample $\sigma_0 = 0.170$. We follow the commonly used MOSI processing protocol from THUIAR releases (Yu et al., 2020; 2021). Video frame embeddings are extracted with OpenFace (Amos et al., 2016). Consistent with prior work, we report F1, PCC, and MAE as evaluation metrics. We include

10 representative baselines. ALMT (Zhang et al., 2023), the leading baseline optimizing with MSE loss, is selected to incorporate the ECA method.

Table 1: Results on three UCI tabular regression tasks. "+ECA" denotes adding ECA onto the FT-Transformer baseline. Rows marked "w/o SRA/DATS/DNPL" are ablation studies that remove the corresponding ECA components. "w/o" = "without" and **bold** indicating best result.

| Method | Appliance | | | Online News | | | Superconductivity | | |
|---|---|---|---|---|---|---|---|---|---|
| | MAE $\downarrow$ | MSE$\times_{10^3} \downarrow$ | PCC $\uparrow$ | MAE $\downarrow$ | MSE$\times_{10^0} \downarrow$ | PCC $\uparrow$ | MAE $\downarrow$ | MSE$\times_{10^2} \downarrow$ | PCC $\uparrow$ |
| FT-Transformer | 39.333 | 6.108 | 0.556 | 0.641 | 0.724 | 0.408 | 8.793 | 1.772 | 0.920 |
| + ECA (full) | **38.665** | **5.790** | **0.598** | **0.631** | **0.712** | **0.420** | **7.976** | **1.582** | **0.930** |
| + ECA (w/o SRA) | 39.208 | 5.994 | 0.575 | 0.637 | 0.725 | 0.410 | 8.377 | 1.695 | 0.920 |
| + ECA (w/o DATS) | 38.906 | 6.037 | 0.561 | 0.645 | 0.740 | 0.418 | 8.630 | 1.709 | 0.927 |
| + ECA (w/o DNPL) | 39.742 | 5.910 | 0.583 | 0.640 | 0.719 | 0.418 | 8.466 | 1.671 | 0.922 |

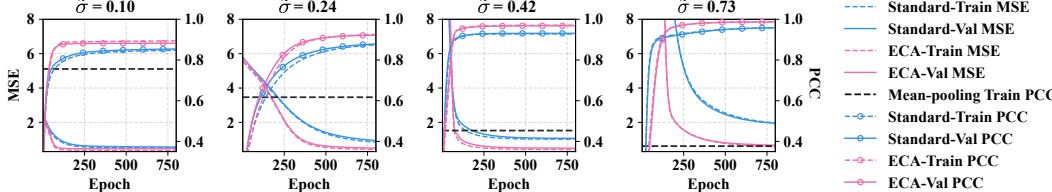

Figure 5: PCC and MSE curves on synthetic datasets with in-sample homogeneity $\tilde{\sigma} \in [0.10, 0.24, 0.42, 0.73]$.

## 5.2 RESULTS AND ANALYSIS

**Synthetic Dataset.** Figure 5 shows case studies on the training and validation curves under different in-sample dispersion (with $\tilde{\sigma}$ = 0.10, 0.24, 0.42, and 0.73). The horizontal line indicates the PCC achieved by mean-pooling over input embeddings. This result confirms our theoretical study in three ways: 1) as the homogeneity intensifies, both the PCC of our ECA and baseline model decrease and the MSE get higher (task harder), and the achievable PCC improvement of convex (*e.g.*, softmax) attention over mean-pooling decreases (Theorem 2.2). 2) ECA consistently outperforms standard attention in both PCC and MSE across all four $\tilde{\sigma}$s, achieving PCC gains of $4.80\%, 5.76\%, 4.68\%, 3.05\%$ and MSE reductions of $20.3\%, 40.8\%, 54.0\%, 66.7\%$ (in order of increasing $\tilde{\sigma}$), showing its ability to explore beyond the convex hull and improving the PCC without compromising the MSE. 3) The PCC curve of ECA keeps improving and converges later and at a higher value than the standard attention baseline, indicating the effectiveness of our proposed DNPL in mitigating the PCC attenuation effect identified in Remark 2.3.

Table 2: Results on MOSI. $\dagger$ from THUIAR's GitHub (Yu et al., 2020; 2021); * from (Hazarika et al., 2020); ** reproduced from public code with provided hyper-parameters.

| Method | F1 $\uparrow$ | MAE $\downarrow$ | PCC $\uparrow$ |
|---|---|---|---|
| TFN$\dagger$ (Zadeh et al., 2017) | 0.791 | 0.947 | 0.673 |
| LMF* (Liu et al., 2018) | 0.824 | 0.917 | 0.695 |
| EF-LSTM$\dagger$ (Williams et al., 2018b) | 0.785 | 0.949 | 0.669 |
| LF-DNN$\dagger$ (Williams et al., 2018a) | 0.786 | 0.955 | 0.658 |
| Graph-MFN$\dagger$ (Zadeh et al., 2018) | 0.784 | 0.956 | 0.649 |
| MulT* (Tsai et al., 2019) | 0.828 | 0.871 | 0.698 |
| MISA$\dagger$ (Hazarika et al., 2020) | 0.836 | 0.777 | 0.778 |
| ICCN* (Sun et al., 2020) | 0.830 | 0.860 | 0.710 |
| DLF** (Wang et al., 2025) | 0.850 | 0.731 | 0.781 |
| ALMT** (Zhang et al., 2023) | 0.851 | 0.721 | 0.783 |
| ALMT+$\mathcal{L}_{PCC}$ | 0.834 | 0.731 | 0.791 |
| ALMT+$\tilde{\mathcal{L}}_{PCC}$ + ECA | **0.859** | **0.695** | **0.806** |

**UCI ML Repository Datasets.** Across three UCI tabular regression tasks, adapting our ECA module into the FT-Transformer yields consistent improvements in both magnitude (MSE and MAE) and shape (PCC) metrics. On Appliance, PCC increases by 0.042 and MSE decreases by $0.318 \times 10^3$; on Online News, PCC increases 0.012 from 0.408 while MSE decreases 0.012 from 0.724; on Superconductivity, PCC $0.920 \rightarrow 0.930$ and MSE $1.772 \times 10^2 \rightarrow 1.582 \times 10^2$. MAE also decreases across all datasets. These improvements provide strong evidence that addressing our identified PCC gradient limitation and the convex attention capacity limit substantially

Table 3: Three-fold regression PCC and MSE on 10xProteomic dataset.

| Method | PCC@F ↑ | PCC@S ↑ | PCC@M ↑ | MSE↓ |
|---|---|---|---|---|
| EGN | $0.602 \pm 0.160$ | $0.647 \pm 0.164$ | $0.629 \pm 0.135$ | $0.056 \pm 0.047$ |
| EGN+ECA | $\mathbf{0.690} \pm 0.202$ | $\mathbf{0.724} \pm 0.191$ | $\mathbf{0.716} \pm 0.168$ | $\mathbf{0.051} \pm 0.048$ |

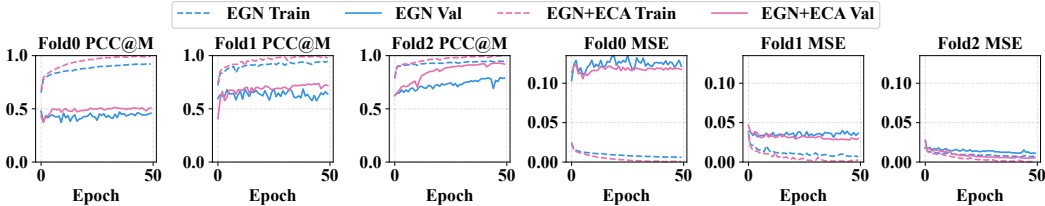

Figure 6: PCC@M and MSE curves for EGN baseline and EGN+ECA on 10xProteomic dataset under each fold. Curves for PCC@S and PCC@F please refer to Figure 8 and Figure 9.

mitigates the PCC plateau without compromising MSE. The ablations further support the contribution of each components: removing DATS lowers Appliance PCC to $0.561$, while removing SRA reduces Online News PCC to $0.418$ and removing DNPL decreases Superconductivity PCC to $0.922$.

**Spatial Transcriptomic Dataset.** We follow the EGN (Yang et al., 2023) setting and report three-fold results. Figure 6 plots PCC and MSE curves for training and validation. Across all folds, integrating ECA consistently improves both metrics. In fold 2, the EGN baseline's PCC flattens near epoch 4, but the MSE keeps decreasing, indicating a clear PCC plateau. In contrast, PCC of EGN+ECA continues to increase, effectively breaking the PCC plateau and improving final validation PCC by $\sim 16.51\%$. The same pattern holds in folds 0 and 1. Throughout training, EGN+ECA also achieves comparable or lower MSE than EGN alone, indicating that ECA successfully preserves magnitude information while achieving better PCC. Table 3 summarizes the overall performance, where EGN+ECA achieves $+14.64\%$ for PCC@F, $+11.89\%$ for PCC@S, $+13.81\%$ for PCC@M, and a $9.83\%$ reduction in MSE, showing that ECA effectively and robustly alleviates the PCC plateau without compromising the MSE.

**Multimodal Sentiment Analysis (MSA) Dataset.** Since ALMT optimizes MSE loss only, we test two settings: (i) ALMT with an additional PCC loss; and (ii) ALMT with ECA adapted into the video attention encoder and trained with the dispersion-normalized PCC loss. Table 2 reports results of 10 baselines. Adding a PCC term yields a small PCC gain but degrades F1 and MAE, reflecting the MSE-PCC conflict under strong in-sample homogeneity. In contrast, adding ECA improves all metrics, achieving a $+2.3\%$ PCC increase without sacrificing F1 or MAE.

## 6 CONCLUSION

This work presents the first theoretical investigation into the PCC plateau phenomenon observed when training attention-based regression models with a joint MSE+PCC loss, particularly under high data homogeneity. Our analysis identified two fundamental limitations in standard softmax attention: conflict in optimization dynamics that attenuate the correlation gradient, and an achievable PCC bound imposed by convex aggregation. To address these bottlenecks, we introduced ECA, a novel plug-in framework incorporating mechanisms to stabilize optimization, adapt to homogeneity, and extrapolate beyond the convex hull. Comprehensive experiments validate our theoretical insights and demonstrate that ECA successfully breaks the PCC plateau, achieving significant correlation gains while maintaining competitive magnitude performance.

## 7 ACKNOWLEDGEMENT

This work was partially supported by US National Science Foundation IIS-2412195, CCF-2400785, the Cancer Prevention and Research Institute of Texas (CPRIT) award (RP230363), the National Institutes of Health (NIH) R01 award (1R01AI190103-01) and Microsoft Accelerate Foundation Models Research (2024).

## 8 REPRODUCIBILITY STATEMENT

We support reproducibility as follows.

**Theoretical Study**   The appendix has complete proofs for all theorems, lemmas, and corollaries in the paper. See Appendix A for results from Section 2.2, Appendix B for Section 2.3, and Appendix D for Section 3.

**Dataset Processing**   Processing details for all datasets (synthetic and real-world) are listed in Appendix G.

**Code Reproducibility**   We include an anonymous zip file with implementations for the synthetic and spatial transcriptomics datasets along with the hyperparameters we use for review. For the spatial transcriptomic dataset, we follow the EGN baseline preprocessing protocol. Due to the limited space, we did not include the dataset. The README.md explains how to get the data and run the code. The complete code can be found in `https://github.com/jyan97/ECA-Extrapolative-Correlation-Attention`.

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

# Appendix

## A PRELIMINARIES

### A.1 PROOF FOR PROPOSITION 2.1

**Proposition A.1** (MSE Mean–std–correlation Decomposition). *Let $y, \hat{y} \in \mathbb{R}$ be the ground truth and predictions across $S$ samples. Let $\mu_y, \mu_{\hat{y}}$ be the empirical means, and $\sigma_y, \sigma_{\hat{y}}$ be the empirical standard deviations. The Mean Squared Error can be decomposed as:*

$$\text{MSE}(y, \hat{y}) = (\mu_{\hat{y}} - \mu_y)^2 + (\sigma_{\hat{y}} - \sigma_y)^2 + 2\, \sigma_y\, \sigma_{\hat{y}}\, (1 - \rho). \tag{13}$$

*Proof.* Write the error for sample $i$ as

$$y_i - \hat{y}_i = \big[(y_i - \mu_y) - (\hat{y}_i - \mu_{\hat{y}})\big] + (\mu_y - \mu_{\hat{y}}). \tag{14}$$

Squaring and averaging over $i = 1, \dots, S$ give us

$$\text{MSE}(y, \hat{y}) = \frac{1}{N} \sum_{i=1}^{N} (y_i - \hat{y}_i)^2 \tag{15}$$

$$= \frac{1}{N} \sum_{i=1}^{N} \Big( (y_i - \mu_y) - (\hat{y}_i - \mu_{\hat{y}}) \Big)^2 + (\mu_y - \mu_{\hat{y}})^2 \tag{16}$$

$$+ \frac{2(\mu_y - \mu_{\hat{y}})}{N} \sum_{i=1}^{N} \Big( (y_i - \mu_y) - (\hat{y}_i - \mu_{\hat{y}}) \Big). \tag{17}$$

The last sum vanishes since $\sum_i (y_i - \mu_y) = \sum_i (\hat{y}_i - \mu_{\hat{y}}) = 0$. Expanding the remaining square and using the definitions of variances, $\text{Cov}(y, \hat{y}) = \sigma_y \sigma_{\hat{y}} \rho$, we obtain

$$\text{MSE}(y, \hat{y}) = (\mu_{\hat{y}} - \mu_y)^2 + \sigma_y^2 + \sigma_{\hat{y}}^2 - \frac{2}{N} \sum_{i=1}^{N} (y_i - \mu_y)(\hat{y}_i - \mu_{\hat{y}}) \tag{18}$$

$$= (\mu_{\hat{y}} - \mu_y)^2 + \sigma_y^2 + \sigma_{\hat{y}}^2 - 2\sigma_y \sigma_{\hat{y}} \rho. \tag{19}$$

Finally, rearranging $\sigma_y^2 + \sigma_{\hat{y}}^2 - 2\, \sigma_y\, \sigma_{\hat{y}}\, \rho = (\sigma_{\hat{y}} - \sigma_y)^2 + 2\, \sigma_y\, \sigma_{\hat{y}}\, (1 - \rho)$ yields the claimed decomposition. $\square$

### A.2 PROOF FOR LEMMA 2.1

**Lemma A.1** (Scaling Invariance of PCC). *Let $m \in \mathbb{R} \setminus \{0\}$ and $n \in \mathbb{R}$. For any sample $\{(y_i, \hat{y}_i)\}_{i=1}^{S}$,*

$$\text{PCC}(y, m\hat{y} + n) = \text{sign}(m)\, \text{PCC}(y, \hat{y}). \tag{20}$$

*Proof.* Using $\rho(u, v) = \frac{\text{Cov}(u,v)}{\sigma_u \sigma_v}$ and $\sigma_{m\hat{y}+n} = |m|\, \sigma_{\hat{y}}$,

$$\text{PCC}(y, m\hat{y} + n) = \frac{\text{Cov}(y, m\hat{y} + n)}{\sigma_y\, \sigma_{m\hat{y}+n}} = \frac{m\, \text{Cov}(y, \hat{y})}{\sigma_y\, |m|\, \sigma_{\hat{y}}} \tag{21}$$

$$= \frac{m}{|m|}\, \text{PCC}(y, \hat{y}) = \text{sign}(m)\, \text{PCC}(y, \hat{y}) \tag{22}$$

$\square$

# B GRADIENT ANALYSIS OF CORRELATION

## B.1 GRADIENT OF PEARSON CORRELATION W.R.T. ATTENTION LOGITS

**Lemma B.1** (Softmax Aggregator Jacobian). *The derivative of the aggregated embedding $\mathbf{v}_s$ with respect to a pre-softmax logit $z_{si}$ is $\partial\mathbf{v}_s/\partial z_{si} = \alpha_{si}(\mathbf{h}_{si} - \mathbf{v}_s)$ and consequently, $\partial\hat{y}_s/\partial z_{si} = \alpha_{si}\mathbf{w}^\top(\mathbf{h}_{si} - \mathbf{v}_s)$.*

*Proof.* Within a fixed sample $s$, the aggregated embedding is $\mathbf{v}_s = \sum_{j=1}^{n_s} \alpha_{sj}\mathbf{h}_{sj}$. We first recall the derivative of the softmax function. The partial derivative of the $j$-th attention weight $\alpha_{sj}$ with respect to the $i$-th input logit $z_{si}$ is given by:

$$\frac{\partial\alpha_{sj}}{\partial z_{si}} = \alpha_{sj}(\delta_{ij} - \alpha_{si}), \tag{23}$$

where $\delta_{ij}$ is the Kronecker delta ($\delta_{ij} = 1$ if $i = j$, and 0 otherwise).

We can now compute the derivative of $\mathbf{v}_s$ with respect to $z_{si}$ using the chain rule:

$$\frac{\partial\mathbf{v}_s}{\partial z_{si}} = \sum_{j=1}^{n_s} \frac{\partial\alpha_{sj}}{\partial z_{si}}\mathbf{h}_{sj} \tag{24}$$

$$= \sum_{j=1}^{n_s} \alpha_{sj}(\delta_{ij} - \alpha_{si})\mathbf{h}_{sj} \tag{25}$$

$$= \sum_{j=1}^{n_s} \alpha_{sj}\delta_{ij}\mathbf{h}_{sj} - \sum_{j=1}^{n_s} \alpha_{sj}\alpha_{si}\mathbf{h}_{sj}. \tag{26}$$

The first term simplifies because $\delta_{ij}$ is non-zero only when $j = i$:

$$\sum_{j=1}^{n_s} \alpha_{sj}\delta_{ij}\mathbf{h}_{sj} = \alpha_{si}\mathbf{h}_{si}. \tag{27}$$

In the second term, $\alpha_{si}$ is independent of the summation index $j$ and can be factored out:

$$\sum_{j=1}^{n_s} \alpha_{sj}\alpha_{si}\mathbf{h}_{sj} = \alpha_{si}\sum_{j=1}^{n_s} \alpha_{sj}\mathbf{h}_{sj} = \alpha_{si}\mathbf{v}_s. \tag{28}$$

Combining these results, we obtain:

$$\frac{\partial\mathbf{v}_s}{\partial z_{si}} = \alpha_{si}\mathbf{h}_{si} - \alpha_{si}\mathbf{v}_s = \alpha_{si}(\mathbf{h}_{si} - \mathbf{v}_s). \tag{29}$$

Consequently, since the prediction is $\hat{y}_s = \mathbf{w}^\top\mathbf{v}_s + c$, its derivative is:

$$\frac{\partial\hat{y}_s}{\partial z_{si}} = \mathbf{w}^\top\frac{\partial\mathbf{v}_s}{\partial z_{si}} = \alpha_{si}\mathbf{w}^\top(\mathbf{h}_{si} - \mathbf{v}_s). \tag{30}$$

$\square$

**Theorem B.1** (Gradient of PCC w.r.t. Attention Logits). *For any $s$, the derivative of Pearson correlation $\rho$ with respect to a pre-softmax logit $z_{si}$ is*

$$\frac{\partial\rho}{\partial z_{si}} = \frac{1}{S\sigma_{\hat{y}}}\left(\frac{a_s}{\sigma_y} - \rho\frac{b_s}{\sigma_{\hat{y}}}\right)\alpha_{si}\mathbf{w}^\top(\mathbf{h}_{si} - \mathbf{v}_s). \tag{31}$$

*Proof.* We express the Pearson correlation $\rho$ as the ratio $\rho = N/D$, where $N$ is the covariance and $D$ is the product of standard deviations.

$$N := \mathrm{Cov}(y, \hat{y}) = \frac{1}{S}\sum_{t=1}^{S} a_t b_t, \quad D := \sigma_y\sigma_{\hat{y}}. \tag{32}$$

Recall that $a_t := y_t - \mu_y$ and $b_t := \hat{y}_t - \mu_{\hat{y}}$ are the centered targets and predictions, respectively, satisfying $\sum_t a_t = 0$ and $\sum_t b_t = 0$.

By the quotient rule, the derivative of $\rho$ is:

$$\frac{\partial \rho}{\partial z_{si}} = \frac{1}{D}\frac{\partial N}{\partial z_{si}} - \frac{N}{D^2}\frac{\partial D}{\partial z_{si}} = \frac{1}{D}\left(\frac{\partial N}{\partial z_{si}} - \rho\frac{\partial D}{\partial z_{si}}\right). \tag{33}$$

We compute the derivatives of $N$ and $D$ separately. Note that the logit $z_{si}$ only directly affects the prediction of sample $s$, i.e., $\partial \hat{y}_t / \partial z_{si} = 0$ if $t \neq s$.

**Step 1: Derivative of the Covariance ($N$).**

$$\frac{\partial N}{\partial z_{si}} = \frac{1}{S}\sum_{t=1}^{S} a_t \frac{\partial b_t}{\partial z_{si}}. \tag{34}$$

We expand the derivative of the centered prediction $b_t = \hat{y}_t - \mu_{\hat{y}}$:

$$\frac{\partial b_t}{\partial z_{si}} = \frac{\partial \hat{y}_t}{\partial z_{si}} - \frac{\partial \mu_{\hat{y}}}{\partial z_{si}} = \frac{\partial \hat{y}_t}{\partial z_{si}} - \frac{1}{S}\sum_{u=1}^{S}\frac{\partial \hat{y}_u}{\partial z_{si}}. \tag{35}$$

Substituting this back into the expression for $\partial N / \partial z_{si}$:

$$\frac{\partial N}{\partial z_{si}} = \frac{1}{S}\sum_t a_t\left(\frac{\partial \hat{y}_t}{\partial z_{si}} - \frac{1}{S}\sum_{u=1}^{S}\frac{\partial \hat{y}_u}{\partial z_{si}}\right) \tag{36}$$

$$= \frac{1}{S}\left(\sum_t a_t\frac{\partial \hat{y}_t}{\partial z_{si}}\right) - \frac{1}{S^2}\left(\sum_t a_t\right)\left(\sum_{u=1}^{S}\frac{\partial \hat{y}_u}{\partial z_{si}}\right). \tag{37}$$

Since the targets are centered ($\sum_t a_t = 0$), the second term vanishes:

$$\frac{\partial N}{\partial z_{si}} = \frac{1}{S}\sum_t a_t\frac{\partial \hat{y}_t}{\partial z_{si}}. \tag{38}$$

Since $\partial \hat{y}_t / \partial z_{si} = 0$ for $t \neq s$, the summation collapses to a single term:

$$\frac{\partial N}{\partial z_{si}} = \frac{1}{S}a_s\frac{\partial \hat{y}_s}{\partial z_{si}}. \tag{39}$$

**Step 2: Derivative of the Standard Deviation Product ($D$).** Since $\sigma_y$ is constant with respect to $z_{si}$, we have $\partial D / \partial z_{si} = \sigma_y(\partial \sigma_{\hat{y}} / \partial z_{si})$. To find the derivative of $\sigma_{\hat{y}}$, we first differentiate the variance $\sigma_{\hat{y}}^2 = \frac{1}{S}\sum_t b_t^2$.

$$\frac{\partial \sigma_{\hat{y}}^2}{\partial z_{si}} = \frac{\partial}{\partial z_{si}}\left(\frac{1}{S}\sum_t b_t^2\right) = \frac{1}{S}\sum_t 2b_t\frac{\partial b_t}{\partial z_{si}}. \tag{40}$$

Similar to Step 1, we substitute the expression for $\partial b_t / \partial z_{si}$ and use the fact that the predictions are centered ($\sum_t b_t = 0$):

$$\frac{\partial \sigma_{\hat{y}}^2}{\partial z_{si}} = \frac{2}{S}\sum_t b_t\left(\frac{\partial \hat{y}_t}{\partial z_{si}} - \frac{1}{S}\sum_{u=1}^{S}\frac{\partial \hat{y}_u}{\partial z_{si}}\right) \tag{41}$$

$$= \frac{2}{S}\left(\sum_t b_t\frac{\partial \hat{y}_t}{\partial z_{si}}\right) - \frac{2}{S^2}\left(\sum_t b_t\right)\left(\sum_{u=1}^{S}\frac{\partial \hat{y}_u}{\partial z_{si}}\right) \tag{42}$$

$$= \frac{2}{S}\sum_t b_t\frac{\partial \hat{y}_t}{\partial z_{si}}. \tag{43}$$

Again, since $\partial \hat{y}_t / \partial z_{si} = 0$ for $t \neq s$:

$$\frac{\partial \sigma_{\hat{y}}^2}{\partial z_{si}} = \frac{2}{S} b_s \frac{\partial \hat{y}_s}{\partial z_{si}}. \tag{44}$$

We now use the chain rule: $\frac{\partial \sigma_{\hat{y}}^2}{\partial z_{si}} = 2\sigma_{\hat{y}} \frac{\partial \sigma_{\hat{y}}}{\partial z_{si}}$. Equating the two expressions and solving for $\frac{\partial \sigma_{\hat{y}}}{\partial z_{si}}$ (assuming $\sigma_{\hat{y}} > 0$):

$$2\sigma_{\hat{y}} \frac{\partial \sigma_{\hat{y}}}{\partial z_{si}} = \frac{2}{S} b_s \frac{\partial \hat{y}_s}{\partial z_{si}} \quad \Longrightarrow \quad \frac{\partial \sigma_{\hat{y}}}{\partial z_{si}} = \frac{b_s}{S\sigma_{\hat{y}}} \frac{\partial \hat{y}_s}{\partial z_{si}}. \tag{45}$$

Therefore, the derivative of the denominator $D$ is:

$$\frac{\partial D}{\partial z_{si}} = \sigma_y \frac{\partial \sigma_{\hat{y}}}{\partial z_{si}} = \frac{\sigma_y b_s}{S\sigma_{\hat{y}}} \frac{\partial \hat{y}_s}{\partial z_{si}}. \tag{46}$$

**Step 3: Combining the results.** We substitute the derivatives of $N$ (Equation (39)) and $D$ (Equation (46)) back into the quotient rule formula (Equation (33)).

$$\frac{\partial \rho}{\partial z_{si}} = \frac{1}{D} \left( \frac{\partial N}{\partial z_{si}} - \rho \frac{\partial D}{\partial z_{si}} \right) \tag{47}$$

$$= \frac{1}{\sigma_y \sigma_{\hat{y}}} \left( \frac{a_s}{S} \frac{\partial \hat{y}_s}{\partial z_{si}} - \rho \frac{\sigma_y b_s}{S\sigma_{\hat{y}}} \frac{\partial \hat{y}_s}{\partial z_{si}} \right) \tag{48}$$

$$= \frac{1}{S\sigma_y \sigma_{\hat{y}}} \left( a_s - \rho \frac{\sigma_y b_s}{\sigma_{\hat{y}}} \right) \frac{\partial \hat{y}_s}{\partial z_{si}} \tag{49}$$

$$= \frac{1}{S\sigma_{\hat{y}}} \left( \frac{a_s}{\sigma_y} - \rho \frac{b_s}{\sigma_{\hat{y}}} \right) \frac{\partial \hat{y}_s}{\partial z_{si}}. \tag{50}$$

Finally, we substitute the expression for $\partial \hat{y}_s / \partial z_{si}$ derived in Lemma B.1 (Equation (30)):

$$\frac{\partial \rho}{\partial z_{si}} = \frac{1}{S\sigma_{\hat{y}}} \left( \frac{a_s}{\sigma_y} - \rho \frac{b_s}{\sigma_{\hat{y}}} \right) \alpha_{si} \mathbf{w}^\top (\mathbf{h}_{si} - \mathbf{v}_s). \tag{51}$$

This concludes the proof. $\qquad \square$

## C  PROOFS FOR OPTIMIZATION DYNAMICS ANALYSIS

### C.1  PROOF OF LEMMA 2.3 (GRADIENT OF MSE W.R.T. ATTENTION LOGITS)

*Proof.* The Mean Squared Error (MSE) loss is defined as:

$$\mathcal{L}_{\mathrm{MSE}} = \frac{1}{S} \sum_{k=1}^{S} (y_k - \hat{y}_k)^2. \tag{52}$$

We compute the derivative with respect to the attention logit $z_{si}$ using the chain rule:

$$\frac{\partial \mathcal{L}_{\mathrm{MSE}}}{\partial z_{si}} = \sum_{k=1}^{S} \frac{\partial \mathcal{L}_{\mathrm{MSE}}}{\partial \hat{y}_k} \frac{\partial \hat{y}_k}{\partial z_{si}}. \tag{53}$$

The derivative of the loss w.r.t. the prediction $\hat{y}_k$ is:

$$\frac{\partial \mathcal{L}_{\mathrm{MSE}}}{\partial \hat{y}_k} = \frac{1}{S} \cdot 2(y_k - \hat{y}_k) \cdot (-1) = \frac{2}{S} (\hat{y}_k - y_k). \tag{54}$$

The derivative of the prediction $\hat{y}_k$ w.r.t. the logit $z_{si}$ is non-zero only if $k = s$. From Lemma 2.2, we have:

$$\frac{\partial \hat{y}_s}{\partial z_{si}} = \alpha_{si} \mathbf{w}^\top (\mathbf{h}_{si} - \mathbf{v}_s). \tag{55}$$

Combining these results:

$$\frac{\partial \mathcal{L}_{\mathrm{MSE}}}{\partial z_{si}} = \frac{2}{S} (\hat{y}_s - y_s) \alpha_{si} \mathbf{w}^\top (\mathbf{h}_{si} - \mathbf{v}_s). \tag{56}$$

$\qquad \square$

## C.2 PROOF OF COROLLARY 2.1 (PCC/MSE GRADIENT RATIO DECAY)

*Proof.* We analyze the ratio of the global scaling factors $g_s^{\text{MSE}}$ and $g_s^{\text{PCC}}$ identified in Section 2.3:

$$g_s^{\text{MSE}} = 2(\hat{y}_s - y_s), \tag{57}$$

$$g_s^{\text{PCC}} = \frac{1}{\sigma_{\hat{y}}} \left( \frac{a_s}{\sigma_y} - \rho \frac{b_s}{\sigma_{\hat{y}}} \right). \tag{58}$$

We analyze their typical scale across samples using their Root Mean Square (RMS) values. We denote the empirical average over the batch as $\mathbb{E}_s[\cdot]$.

**Step 1: RMS scale of the PCC global factor.** We define normalized variables $A_s := a_s/\sigma_y$ and $B_s := b_s/\sigma_{\hat{y}}$. By construction, $A_s$ and $B_s$ have zero mean and unit variance ($\text{Var}(A_s) = \text{Var}(B_s) = 1$), and their covariance is the PCC ($\text{Cov}(A_s, B_s) = \rho$). The term inside the parenthesis of $g_s^{\text{PCC}}$ is $A_s - \rho B_s$. Its variance is:

$$\text{Var}(A_s - \rho B_s) = \text{Var}(A_s) + \rho^2 \text{Var}(B_s) - 2\rho \, \text{Cov}(A_s, B_s) \tag{59}$$

$$= 1 + \rho^2 - 2\rho^2 = 1 - \rho^2. \tag{60}$$

Since the mean of $A_s - \rho B_s$ is zero, the RMS magnitude of $g_s^{\text{PCC}}$ across samples is:

$$\text{RMS}_s(g_s^{\text{PCC}}) = \sqrt{\mathbb{E}_s[(g_s^{\text{PCC}})^2]} = \frac{1}{\sigma_{\hat{y}}} \sqrt{\text{Var}(A_s - \rho B_s)} = \frac{\sqrt{1 - \rho^2}}{\sigma_{\hat{y}}}. \tag{61}$$

**Step 2: RMS scale of the MSE global factor.** For the MSE global factor, we have:

$$\text{RMS}_s(g_s^{\text{MSE}}) = \sqrt{\mathbb{E}_s[[2(\hat{y}_s - y_s)]^2]} = 2\sqrt{\mathbb{E}_s[(\hat{y}_s - y_s)^2]} \tag{62}$$

$$= 2\sqrt{\text{MSE}(\mathbf{y}, \hat{\mathbf{y}})}. \tag{63}$$

**Step 3: Bounding the ratio of RMS global factors.** The ratio $r_{\text{global}}$ is defined as:

$$r_{\text{global}} = \frac{\text{RMS}_s(g_s^{\text{PCC}})}{\text{RMS}_s(g_s^{\text{MSE}})} = \frac{\sqrt{1 - \rho^2}}{2\sigma_{\hat{y}}\sqrt{\text{MSE}}}. \tag{64}$$

We use the MSE decomposition (Theorem 2.1):

$$\text{MSE}(\mathbf{y}, \hat{\mathbf{y}}) = (\mu_{\hat{\mathbf{y}}} - \mu_{\mathbf{y}})^2 + (\sigma_{\hat{\mathbf{y}}} - \sigma_{\mathbf{y}})^2 + 2\, \sigma_{\mathbf{y}}\sigma_{\hat{\mathbf{y}}}(1 - \rho). \tag{65}$$

Since all terms are non-negative, we obtain a lower bound for MSE:

$$\text{MSE}(\mathbf{y}, \hat{\mathbf{y}}) \geq 2\, \sigma_y \, \sigma_{\hat{y}}(1 - \rho). \tag{66}$$

We assume $\rho \in [0, 1]$, which is typical during training. We use the inequality:

$$1 - \rho^2 = (1 - \rho)(1 + \rho) \leq 2(1 - \rho) \implies \sqrt{1 - \rho^2} \leq \sqrt{2}\sqrt{1 - \rho}. \tag{67}$$

Plugging the bounds from equation 66 and equation 67 into the ratio definition equation 64:

$$r_{\text{global}} \leq \frac{\sqrt{2}\sqrt{1 - \rho}}{2\sigma_{\hat{y}}\sqrt{2\sigma_y\sigma_{\hat{y}}(1 - \rho)}} \tag{68}$$

$$= \frac{\sqrt{2}\sqrt{1 - \rho}}{2\sigma_{\hat{y}} \cdot \sqrt{2}\sqrt{\sigma_y\sigma_{\hat{y}}}\sqrt{1 - \rho}} \tag{69}$$

$$= \frac{1}{2\sigma_{\hat{y}}\sqrt{\sigma_y\sigma_{\hat{y}}}} = \frac{1}{2\sqrt{\sigma_y}} \cdot \frac{1}{\sigma_{\hat{y}}^{3/2}}. \tag{70}$$

This completes the proof. $\qquad\square$

### C.3 DERIVATION OF THE GRADIENT MAGNITUDE BOUND

**Lemma C.1** (Within-sample Dispersion Bound). *Recall the definitions $\boldsymbol{\mu}_s = \frac{1}{n_s} \sum_{j=1}^{n_s} \mathbf{h}_{sj}$ (within-sample mean) and $\sigma_s^2 = \frac{1}{n_s} \sum_{j=1}^{n_s} \|\mathbf{h}_{sj} - \boldsymbol{\mu}_s\|^2$ (within-sample variance). Also recall $\mathbf{v}_s = \sum_{j=1}^{n_s} \alpha_{sj} \mathbf{h}_{sj}$ where $\alpha_{sj} \geq 0$ and $\sum_j \alpha_{sj} = 1$. Then for every $i \in \{1, \dots, n_s\}$,*

$$\|\mathbf{h}_{si} - \mathbf{v}_s\| \leq 2\sqrt{n_s}\, \sigma_s.$$

*Proof.* Fix sample $s$ and index $i$. We use the triangle inequality by inserting the within-sample mean $\boldsymbol{\mu}_s$:

$$\|\mathbf{h}_{si} - \mathbf{v}_s\| = \|\mathbf{h}_{si} - \boldsymbol{\mu}_s + \boldsymbol{\mu}_s - \mathbf{v}_s\| \leq \|\mathbf{h}_{si} - \boldsymbol{\mu}_s\| + \|\mathbf{v}_s - \boldsymbol{\mu}_s\|. \tag{71}$$

We bound each term on the right-hand side separately.

**Term 1:** $\|\mathbf{h}_{si} - \boldsymbol{\mu}_s\|$. We first bound the deviation by the maximum deviation within the sample:

$$\|\mathbf{h}_{si} - \boldsymbol{\mu}_s\| \leq \max_j \|\mathbf{h}_{sj} - \boldsymbol{\mu}_s\|. \tag{72}$$

The maximum of non-negative numbers is bounded by the square root of the sum of their squares (i.e., $x_k^2 \leq \sum_j x_j^2$ implies $x_k \leq \sqrt{\sum_j x_j^2}$):

$$\max_j \|\mathbf{h}_{sj} - \boldsymbol{\mu}_s\| \leq \sqrt{\sum_{j=1}^{n_s} \|\mathbf{h}_{sj} - \boldsymbol{\mu}_s\|^2}. \tag{73}$$

By the definition of $\sigma_s^2$, the sum of squares is $n_s \sigma_s^2$. Thus,

$$\|\mathbf{h}_{si} - \boldsymbol{\mu}_s\| \leq \sqrt{n_s \sigma_s^2} = \sqrt{n_s}\, \sigma_s. \tag{74}$$

**Term 2:** $\|\mathbf{v}_s - \boldsymbol{\mu}_s\|$. We express the deviation of the aggregated embedding $\mathbf{v}_s$ from the mean $\boldsymbol{\mu}_s$. Since $\sum_j \alpha_{sj} = 1$, we have $\boldsymbol{\mu}_s = \sum_j \alpha_{sj} \boldsymbol{\mu}_s$.

$$\mathbf{v}_s - \boldsymbol{\mu}_s = \sum_{j=1}^{n_s} \alpha_{sj} \mathbf{h}_{sj} - \sum_{j=1}^{n_s} \alpha_{sj} \boldsymbol{\mu}_s = \sum_{j=1}^{n_s} \alpha_{sj} (\mathbf{h}_{sj} - \boldsymbol{\mu}_s). \tag{75}$$

Using the convexity of the norm (Jensen's inequality):

$$\|\mathbf{v}_s - \boldsymbol{\mu}_s\| = \left\| \sum_{j=1}^{n_s} \alpha_{sj} (\mathbf{h}_{sj} - \boldsymbol{\mu}_s) \right\| \leq \sum_{j=1}^{n_s} \alpha_{sj} \|\mathbf{h}_{sj} - \boldsymbol{\mu}_s\|. \tag{76}$$

This weighted average is bounded by the maximum element:

$$\sum_{j=1}^{n_s} \alpha_{sj} \|\mathbf{h}_{sj} - \boldsymbol{\mu}_s\| \leq \max_j \|\mathbf{h}_{sj} - \boldsymbol{\mu}_s\|. \tag{77}$$

As established for Term 1 (Equation (74)), the maximum deviation is bounded by $\sqrt{n_s}\, \sigma_s$. Therefore,

$$\|\mathbf{v}_s - \boldsymbol{\mu}_s\| \leq \sqrt{n_s}\, \sigma_s. \tag{78}$$

**Conclusion.** Substituting the bounds from Equation (74) and Equation (78) into Equation (71):

$$\|\mathbf{h}_{si} - \mathbf{v}_s\| \leq \sqrt{n_s}\, \sigma_s + \sqrt{n_s}\, \sigma_s = 2\sqrt{n_s}\, \sigma_s. \qquad \square$$

**Lemma C.2** (Magnitude Bound of a Centered, Unit-variance Vector). *Let $x_1, \dots, x_S \in \mathbb{R}$ satisfy $\sum_{s=1}^{S} x_s = 0$ and $\frac{1}{S} \sum_{s=1}^{S} x_s^2 = 1$ (equivalently $\sum_{s=1}^{S} x_s^2 = S$). Then for every $j \in \{1, \dots, S\}$, we have $|x_j| \leq \sqrt{S-1}$.*

*Proof.* Fix an index $j$. Since the vector is centered ($\sum_{s=1}^{S} x_s = 0$), we can express $x_j$ in terms of the other elements: $x_j = -\sum_{s \neq j} x_s$. We analyze the squared magnitude $x_j^2$ using the Cauchy–Schwarz inequality and view the summation as a dot product between a vector of ones $\mathbf{1} \in \mathbb{R}^{S-1}$ and the vector $(x_s)_{s \neq j} \in \mathbb{R}^{S-1}$.

$$x_j^2 = \left(\sum_{s \neq j} 1 \cdot x_s\right)^2 \leq \left(\sum_{s \neq j} 1^2\right)\left(\sum_{s \neq j} x_s^2\right) = (S-1)\sum_{s \neq j} x_s^2. \tag{79}$$

We use the unit-variance condition, $\sum_{s=1}^{S} x_s^2 = S$. Therefore, $\sum_{s \neq j} x_s^2 = S - x_j^2$. Substituting this into the inequality:

$$x_j^2 \leq (S-1)\left(S - x_j^2\right) = S(S-1) - (S-1)x_j^2. \tag{80}$$

Rearranging the terms to isolate $x_j^2$:

$$x_j^2 + (S-1)x_j^2 \leq S(S-1) \tag{81}$$
$$S\, x_j^2 \leq S(S-1). \tag{82}$$

Dividing by $S$ (which is positive) gives $x_j^2 \leq S-1$. Taking the square root yields the desired bound:

$$|x_j| \leq \sqrt{S-1}. \qquad \square$$

**Corollary C.1** (Gradient Magnitude Bound). *The magnitude of the PCC gradient in Theorem 2.1 can be bounded by*

$$\left|\frac{\partial \rho}{\partial z_{si}}\right| \leq \frac{1}{\sigma_{\hat{y}}} \frac{4\sqrt{n_s(S-1)}}{S} \|\mathbf{w}\| \sigma_s. \tag{83}$$

*Proof.* We start from the expression for the PCC gradient derived in Theorem B.1:

$$\frac{\partial \rho}{\partial z_{si}} = \frac{1}{S\sigma_{\hat{y}}} \left(\frac{a_s}{\sigma_y} - \rho\frac{b_s}{\sigma_{\hat{y}}}\right) \alpha_{si}\mathbf{w}^\top(\mathbf{h}_{si} - \mathbf{v}_s). \tag{84}$$

We analyze the magnitude of this expression by applying the triangle inequality and the Cauchy-Schwarz inequality ($|\mathbf{w}^\top\mathbf{x}| \leq \|\mathbf{w}\|\|\mathbf{x}\|$):

$$\left|\frac{\partial \rho}{\partial z_{si}}\right| \leq \frac{|\alpha_{si}|}{S\sigma_{\hat{y}}} \left|\frac{a_s}{\sigma_y} - \rho\frac{b_s}{\sigma_{\hat{y}}}\right| \left|\mathbf{w}^\top(\mathbf{h}_{si} - \mathbf{v}_s)\right| \tag{85}$$

$$\leq \frac{|\alpha_{si}|}{S\sigma_{\hat{y}}} \left(\left|\frac{a_s}{\sigma_y}\right| + |\rho|\left|\frac{b_s}{\sigma_{\hat{y}}}\right|\right) \|\mathbf{w}\|\,\|\mathbf{h}_{si} - \mathbf{v}_s\|. \tag{86}$$

We now bound the individual components.

1. **Attention weight:** Since $\alpha_s$ is a probability from Softmax, $0 \leq \alpha_{si} \leq 1$

2. **Correlation coefficient:** By definition, $-1 \leq \rho \leq 1$, so $|\rho| \leq 1$

3. **Standardized scores:** The terms $\frac{a_s}{\sigma_y}$ and $\frac{b_s}{\sigma_{\hat{y}}}$ are the standardized scores (z-scores) of the target and prediction for sample $s$. They form centered, unit-variance vectors across the $S$ samples. By applying Lemma C.2, we have:

$$\left|\frac{a_s}{\sigma_y}\right| \leq \sqrt{S-1} \quad \text{and} \quad \left|\frac{b_s}{\sigma_{\hat{y}}}\right| \leq \sqrt{S-1}.$$

4. **Within-sample dispersion:** The term $\|\mathbf{h}_{si} - \mathbf{v}_s\|$ represents the deviation of the embedding $\mathbf{h}_{si}$ from the aggregated embedding $\mathbf{v}_s$. By applying Lemma C.1, we have:

$$\|\mathbf{h}_{si} - \mathbf{v}_s\| \leq 2\sqrt{n_s}\,\sigma_s.$$

Substituting these bounds into the inequality:

$$\left| \frac{\partial \rho}{\partial z_{si}} \right| \le \frac{1}{S\sigma_{\hat{y}}} \left( \sqrt{S-1} + 1\sqrt{S-1} \right) \|\mathbf{w}\| \left( 2\sqrt{n_s}\sigma_s \right) \tag{87}$$

$$= \frac{1}{S\sigma_{\hat{y}}} \left( 2\sqrt{S-1} \right) \|\mathbf{w}\| \left( 2\sqrt{n_s}\sigma_s \right) \tag{88}$$

$$= \frac{4\sqrt{n_s(S-1)}}{S\sigma_{\hat{y}}} \|\mathbf{w}\| \sigma_s. \tag{89}$$

Rearranging the terms to highlight the key factors identified in the main text:

$$\left| \frac{\partial \rho}{\partial z_{si}} \right| \le \underbrace{\frac{1}{\sigma_{\hat{y}}}}_{\substack{\text{prediction} \\ \text{deviation}}} \cdot \underbrace{\frac{4\sqrt{n_s(S-1)}}{S}}_{\text{batch scale}} \cdot \underbrace{\|\mathbf{w}\|}_{\substack{\text{regression} \\ \text{weights}}} \cdot \underbrace{\sigma_s}_{\substack{\text{in-sample} \\ \text{dispersion}}}. \tag{90}$$

This concludes the proof. □

## D    ACHIEVABLE PCC BOUND OF CONVEX ATTENTION MODELS

This section provides the detailed analysis and proof of an intrinsic upper bound on the PCC gain that any convex attention mechanism can achieve compared to a simple mean-pooling baseline. This analysis formalizes the capacity limitation imposed by the convex hull constraint.

**Setup and Decomposition.**    We consider a dataset of $S$ samples. For sample $s$, we have element embeddings $\mathbf{h}_s = \{\mathbf{h}_{si}\}_{i=1}^{n_s}$ where $\mathbf{h}_{si} \in \mathbb{R}^d$. Let $\boldsymbol{\mu}_s := \frac{1}{n_s} \sum_{i=1}^{n_s} \mathbf{h}_{si}$ be the mean-pooling embedding. A convex attention mechanism computes weights $\{\alpha_{si}\}_{i=1}^{n_s}$ such that $\alpha_{si} \ge 0$ and $\sum_i \alpha_{si} = 1$. The aggregated embedding is $\mathbf{v}_s = \sum_i \alpha_{si}\mathbf{h}_{si}$. The prediction is given by a linear regression head $(\mathbf{w}, b)$: $\hat{y}_s = \mathbf{w}^\top \mathbf{v}_s + c$.

We decompose the prediction relative to the mean-pooling baseline:

$$\hat{y}_s = \underbrace{(\mathbf{w}^\top \boldsymbol{\mu}_s + c)}_{\bar{y}_s} + \underbrace{\mathbf{w}^\top (\mathbf{v}_s - \boldsymbol{\mu}_s)}_{\Delta\hat{y}_s}. \tag{91}$$

Here, $\bar{y}_s$ is the baseline prediction, and $\Delta\hat{y}_s$ is the attention-induced perturbation.

**Quantifying Dispersion and Variation.**    We introduce measures for within-sample dispersion (related to in-sample homogeneity) and across-sample variation. Throughout this section, $\|\cdot\|_2$ denotes the $\ell_2$ norm. We assume the empirical definition for standard deviation (normalized by $1/S$).

**Definition D.1** (Intrinsic Dispersion and Baseline Variation). *For each sample $s$, define the maximum within-sample deviation (the radius of the convex hull centered at the mean):*

$$R_s := \max_{1 \le i \le n_s} \big\| \mathbf{h}_{si} - \boldsymbol{\mu}_s \big\|_2. \tag{92}$$

*Define the intrinsic within-sample dispersion $\tilde{R}$ as the root mean square (RMS) of these radii across the dataset:*

$$\tilde{R} := \sqrt{\frac{1}{S} \sum_{s=1}^{S} R_s^2}. \tag{93}$$

*Let $\sigma_0 := \mathrm{std}_s(\bar{y}_s)$ denote the standard deviation of the baseline predictions across samples.*

**Remark D.1.** *$\tilde{R}$ measures the intrinsic homogeneity of the embeddings within samples, independent of the regression head $\mathbf{w}$. $\sigma_0$ captures the variation of the mean embeddings projected onto the regression space. By the scaling property of the standard deviation, $\sigma_0 = \|\mathbf{w}\|_2 \, \mathrm{std}_s(\hat{\mathbf{w}}^\top \boldsymbol{\mu}_s)$, where $\hat{\mathbf{w}} := \mathbf{w}/\|\mathbf{w}\|_2$.*

**Centered Notation.** We define the vectors of predictions across the $S$ samples: $\hat{\mathbf{y}}, \bar{\mathbf{y}}, \Delta\hat{\mathbf{y}} \in \mathbb{R}^S$. From Equation (91), we have $\hat{\mathbf{y}} = \bar{\mathbf{y}} + \Delta\hat{\mathbf{y}}$.

We denote the centered versions of these vectors (by subtracting their respective means $\mu_{\hat{y}}, \mu_{\bar{y}}, \mu_{\Delta\hat{y}}$) as $\mathbf{b}, \bar{\mathbf{b}}, \Delta\mathbf{b}$. By linearity, the decomposition holds for centered vectors: $\mathbf{b} = \bar{\mathbf{b}} + \Delta\mathbf{b}$. We denote the centered ground-truth targets as $\mathbf{a}$, where $a_s = y_s - \mu_y$.

The Pearson correlation coefficient (PCC) is the cosine similarity between centered vectors. Let $\rho = \mathrm{PCC}(\mathbf{a}, \mathbf{b})$ and $\rho_0 = \mathrm{PCC}(\mathbf{a}, \bar{\mathbf{b}})$.

**Bounding the Attention Perturbation.** We first establish bounds on the magnitude of the perturbation $\Delta\hat{\mathbf{y}}$ and its centered counterpart $\Delta\mathbf{b}$.

**Lemma D.1** (Bound on Prediction Perturbation). *For any convex attention weights $\{\alpha_{si}\}$, the perturbation for sample $s$ is bounded by:*

$$|\Delta\hat{y}_s| \leq \|\mathbf{w}\|_2\, R_s.$$

*Consequently, the L2 norm of the centered perturbation vector is bounded by:*

$$\|\Delta\mathbf{b}\|_2 \ \leq\ \sqrt{S}\,\|\mathbf{w}\|_2\,\tilde{R}.$$

*Proof.* We analyze the perturbation term. Since $\sum_i \alpha_{si} = 1$, we can write $\boldsymbol{\mu}_s = \sum_i \alpha_{si}\boldsymbol{\mu}_s$.

$$\Delta\hat{y}_s = \mathbf{w}^\top(\mathbf{v}_s - \boldsymbol{\mu}_s) = \mathbf{w}^\top\left(\sum_{i=1}^{n_s}\alpha_{si}\mathbf{h}_{si} - \sum_{i=1}^{n_s}\alpha_{si}\boldsymbol{\mu}_s\right) = \mathbf{w}^\top\sum_{i=1}^{n_s}\alpha_{si}(\mathbf{h}_{si} - \boldsymbol{\mu}_s).$$

Taking the absolute value:

$$|\Delta\hat{y}_s| \leq \|\mathbf{w}\|_2\left\|\sum_i \alpha_{si}(\mathbf{h}_{si} - \boldsymbol{\mu}_s)\right\| \quad \text{(Cauchy–Schwarz)}$$

$$\leq \|\mathbf{w}\|_2\sum_i \alpha_{si}\|\mathbf{h}_{si} - \boldsymbol{\mu}_s\| \quad\quad \text{(Convexity of norm / Triangle inequality)}$$

$$\leq \|\mathbf{w}\|_2\sum_i \alpha_{si}R_s \quad\quad\quad \text{(Definition of $R_s$)}$$

$$= \|\mathbf{w}\|_2\,R_s.$$

To bound the L2 norm of the uncentered perturbation vector $\Delta\hat{\mathbf{y}}$, we sum the squared bounds across samples:

$$\|\Delta\hat{\mathbf{y}}\|_2^2 = \sum_{s=1}^{S}|\Delta\hat{y}_s|^2 \leq \sum_{s=1}^{S}(\|\mathbf{w}\|_2 R_s)^2 = \|\mathbf{w}\|_2^2\sum_{s=1}^{S}R_s^2.$$

Using the definition of the RMS dispersion $\tilde{R}^2 = \frac{1}{S}\sum_s R_s^2$, we get:

$$\|\Delta\hat{\mathbf{y}}\|_2^2 \leq S\,\|\mathbf{w}\|_2^2\,\tilde{R}^2 \implies \|\Delta\hat{\mathbf{y}}\|_2 \leq \sqrt{S}\,\|\mathbf{w}\|_2\,\tilde{R}.$$

Finally, $\Delta\mathbf{b}$ is the centered vector of $\Delta\hat{\mathbf{y}}$. Centering a vector (projecting onto the subspace orthogonal to the constant vector (Arefidamghani et al., 2022; Wang et al., 2010)) is a non-expansive operation on $\ell_2$ space, hence $\|\Delta\mathbf{b}\|_2 \leq \|\Delta\hat{\mathbf{y}}\|_2 \leq \sqrt{S}\,\|\mathbf{w}\|_2\,\tilde{R}$. $\qquad\square$

**General Correlation Perturbation Lemma.** We utilize a general result bounding the change in the cosine similarity when one vector is perturbed.

**Lemma D.2** (Correlation perturbation). *For any $\mathbf{a}, \mathbf{b}, \boldsymbol{\delta} \in \mathbb{R}^S$ with $\|\boldsymbol{\delta}\|_2 < \|\mathbf{b}\|_2$,*

$$\left|\frac{\mathbf{a}\cdot(\mathbf{b}+\boldsymbol{\delta})}{\|\mathbf{a}\|_2\,\|\mathbf{b}+\boldsymbol{\delta}\|_2} - \frac{\mathbf{a}\cdot\mathbf{b}}{\|\mathbf{a}\|_2\,\|\mathbf{b}\|_2}\right| \ \leq\ \frac{2\,\|\boldsymbol{\delta}\|_2}{\|\mathbf{b}\|_2 - \|\boldsymbol{\delta}\|_2}.$$

*Proof.* Let $\hat{\mathbf{a}} = \mathbf{a}/\|\mathbf{a}\|_2$. Using triangle inequality, we have:

$$\left|\frac{\hat{\mathbf{a}}\cdot(\mathbf{b}+\boldsymbol{\delta})}{\|\mathbf{b}+\boldsymbol{\delta}\|_2} - \frac{\hat{\mathbf{a}}\cdot\mathbf{b}}{\|\mathbf{b}\|_2}\right| = \left|\frac{\hat{\mathbf{a}}\cdot\mathbf{b}}{\|\mathbf{b}+\boldsymbol{\delta}\|_2} - \frac{\hat{\mathbf{a}}\cdot\mathbf{b}}{\|\mathbf{b}\|_2} + \frac{\hat{\mathbf{a}}\cdot\boldsymbol{\delta}}{\|\mathbf{b}+\boldsymbol{\delta}\|_2}\right| \tag{94}$$

$$\leq \underbrace{\left| (\hat{\mathbf{a}} \cdot \mathbf{b}) \left( \frac{1}{|\mathbf{b} + \boldsymbol{\delta}|_2} - \frac{1}{|\mathbf{b}|_2} \right) \right|}_{T_1} + \underbrace{\left| \frac{\hat{\mathbf{a}} \cdot \boldsymbol{\delta}}{|\mathbf{b} + \boldsymbol{\delta}|_2} \right|}_{T_2}. \tag{95}$$

We analyze each term. For $T_2$, by Cauchy–Schwarz ($|\hat{\mathbf{a}} \cdot \boldsymbol{\delta}| \leq \|\boldsymbol{\delta}\|_2$) and the reverse triangle inequality ($\|\mathbf{b} + \boldsymbol{\delta}\|_2 \geq \|\mathbf{b}\|_2 - \|\boldsymbol{\delta}\|_2$), we have $T_2 \leq \frac{\|\boldsymbol{\delta}\|_2}{\|\mathbf{b}\|_2 - \|\boldsymbol{\delta}\|_2}$.

For $T_1$, we rewrite the expression: $T_1 = |\hat{\mathbf{a}} \cdot \mathbf{b}| \left| \frac{\|\mathbf{b}\|_2 - \|\mathbf{b} + \boldsymbol{\delta}\|_2}{\|\mathbf{b}\|_2 \|\mathbf{b} + \boldsymbol{\delta}\|_2} \right|$.

Using Cauchy–Schwarz ($|\hat{\mathbf{a}} \cdot \mathbf{b}| \leq \|\mathbf{b}\|_2$), and the reverse triangle inequality ($\left| \|\mathbf{b}\|_2 - \|\mathbf{b} + \boldsymbol{\delta}\|_2 \right| \leq \|\boldsymbol{\delta}\|_2$) again, we get: $T_1 \leq \|\mathbf{b}\|_2 \frac{\|\boldsymbol{\delta}\|_2}{\|\mathbf{b}\|_2(\|\mathbf{b}\|_2 - \|\boldsymbol{\delta}\|_2)} = \frac{\|\boldsymbol{\delta}\|_2}{\|\mathbf{b}\|_2 - \|\boldsymbol{\delta}\|_2}$. Summing $T_1$ and $T_2$ completes the proof.

$\square$

## D.1 MAIN RESULT: ACHIEVABLE PCC GAIN

We now combine these lemmas to prove the main theorem regarding the capacity ceiling of convex attention.

**Theorem D.1** (Achievable PCC Gain Bound for Convex Attention). *Let $\rho_0$ be the PCC achieved by the mean-pooling baseline $\bar{\mathbf{y}}$, and $\rho$ the PCC achieved by any convex attention mechanism $\hat{\mathbf{y}}$. Assume $\|\mathbf{w}\|_2 > 0$. If the intrinsic dispersion is small relative to the baseline variation such that $\tilde{R} < \sigma_0/\|\mathbf{w}\|_2$, then:*

$$|\rho - \rho_0| \leq \frac{2\|\mathbf{w}\|_2 \tilde{R}}{\sigma_0 - \|\mathbf{w}\|_2 \tilde{R}} = \frac{2\tilde{R}}{\sigma_0/\|\mathbf{w}\|_2 - \tilde{R}}.$$

*Proof.* We apply Lemma D.2 using the centered vectors: $\mathbf{m} = \mathbf{a}$ (centered targets), $\mathbf{n} = \bar{\mathbf{b}}$ (centered baseline predictions), and $\boldsymbol{\delta} = \Delta\mathbf{b}$ (centered attention perturbation).

First, we relate the L2 norms of the centered vectors to the defined dispersion measures. By the definition of the empirical standard deviation $\sigma_0$, we have:

$$\|\bar{\mathbf{b}}\|_2 = \sqrt{\sum_s (\bar{y}_s - \mu_{\bar{y}})^2} = \sqrt{S}\,\sigma_0.$$

By Lemma D.1, the perturbation is bounded by:

$$\|\Delta\mathbf{b}\|_2 \leq \sqrt{S}\,\|\mathbf{w}\|_2 \tilde{R}.$$

Next, we verify the condition required for Lemma D.2, $\|\boldsymbol{\delta}\|_2 < \|\mathbf{n}\|_2$. The assumption $\tilde{R} < \sigma_0/\|\mathbf{w}\|_2$ implies $\|\mathbf{w}\|_2 \tilde{R} < \sigma_0$. Multiplying by $\sqrt{S}$, we get:

$$\sqrt{S}\,\|\mathbf{w}\|_2 \tilde{R} < \sqrt{S}\,\sigma_0.$$

Therefore, $\|\Delta\mathbf{b}\|_2 < \|\bar{\mathbf{b}}\|_2$, satisfying the prerequisite.

Now, applying Lemma D.2:

$$|\rho - \rho_0| \leq \frac{2\|\Delta\mathbf{b}\|_2}{\|\bar{\mathbf{b}}\|_2 - \|\Delta\mathbf{b}\|_2}.$$

Since the function $f(x) = 2x/(C - x)$ is monotonically increasing for $x < C$ (where $C = \|\bar{\mathbf{b}}\|_2$), we substitute the upper bound for $\|\Delta\mathbf{b}\|_2$:

$$|\rho - \rho_0| \leq \frac{2(\sqrt{S}\,\|\mathbf{w}\|_2 \tilde{R})}{\sqrt{S}\,\sigma_0 - (\sqrt{S}\,\|\mathbf{w}\|_2 \tilde{R})}$$

$$= \frac{2\|\mathbf{w}\|_2 \tilde{R}}{\sigma_0 - \|\mathbf{w}\|_2 \tilde{R}}.$$

Dividing the numerator and denominator by $\|\mathbf{w}\|_2$ (which is positive by assumption) yields the scale-invariant form:

$$|\rho - \rho_0| \leq \frac{2\tilde{R}}{\sigma_0/\|\mathbf{w}\|_2 - \tilde{R}}.$$

$\square$

## D.2 CONNECTION BETWEEN IN-SAMPLE DISPERSION AND RADIUS

**Lemma D.3** (Dispersion–Radius Connection). *For each sample $s$, $R_s/\sqrt{n_s} \leq \sigma_s \leq R_s$. Consequently, we denote $\tilde{\sigma} := \left(\frac{1}{S}\sum_{s=1}^{S}\sigma_s^2\right)^{1/2}$ with $n_{\max} = \max_s n_s$, we have $\tilde{R}/\sqrt{n_{\max}} \leq \tilde{\sigma} \leq \tilde{R}$.*

# E GENERALITY OF THEORETICAL ANALYSIS REGARDING ARCHITECTURE DEPTH

A potential concern regarding our theoretical analysis is whether the findings derived for attention aggregation apply to deep, multi-layer architectures. In this section, we clarify that our analysis focuses on the aggregation mechanism at the readout stage, which dictates the final prediction behavior regardless of the depth of the preceding backbone.

**Backbone-Agnostic Formulation.** Our theoretical model assumes input embeddings $\mathbf{h}_s = \{\mathbf{h}_{si}\}_{i=1}^{n_s}$. In the context of deep learning, these are not raw inputs but rather the latent representations produced by a backbone function $f_\theta(\cdot)$ (comprising multiple Transformer blocks, FFNs, residual connections, and LayerNorms). The final prediction is modeled as $\hat{y}_s = \mathbf{w}^\top \mathbf{v}_s + c$, where $\mathbf{v}_s$ is a convex combination of these final-layer representations. This formulation exactly matches the standard architectural paradigm used in modern Transformers:

- **[CLS] Token Aggregation:** In models like BERT or ViT, the prediction is often derived from a specific [CLS] token. However, in a standard Transformer layer, the output embedding of the [CLS] token is computed via the attention mechanism, which is a convex combination (softmax) of the element embeddings from the previous layer. Thus, the readout remains a convex aggregation of the backbone's features, subject to the analysis we present.

- **Global Average Pooling:** Many regression heads utilize global average pooling over token embeddings, which is a special case of convex attention where uniform weights are applied ($\alpha_{si} = 1/n_s$).

**The Readout Bottleneck.** Our analysis (Theorem 2.2) establishes that the achievable PCC gain is bounded by the convex hull of the input embeddings. Even with a highly expressive deep backbone $f_\theta$, if the final aggregation step is convex, the prediction $\hat{y}_s$ is geometrically constrained to the interior of the simplex formed by the final-layer features $\{\mathbf{h}_{si}\}$. Crucially, deep backbones do not inherently solve the *homogeneity* issue. In fact, models pre-trained with contrastive objectives often produce highly homogeneous embeddings for in-sample elements (semantically similar tokens or patches), which shrinks the convex hull radius $R_s$ and exacerbates the capacity limitation.

**Empirical Validation on Deep Architectures.** Our experimental evaluation in Section 5.1 and Table 2 utilizes deep, multi-layer architectures, not shallow regressors.

- **FT-Transformer:** A deep tabular Transformer with multiple attention layers.

- **ALMT:** A complex multimodal Transformer for sentiment analysis.

- **EGN:** A multi-layer vision Transformer model for spatial transcriptomics.

In all these deep settings, we observed the PCC plateau phenomenon. The consistent improvement provided by ECA confirms that the convex readout mechanism acts as a bottleneck even in deep models, and that relaxing this constraint via extrapolative mechanisms is necessary for optimal correlation learning.

# F   EXTENDED EMPIRICAL EVIDENCE OF THE PCC PLATEAU

To rigorously address the universality of the PCC plateau phenomenon and demonstrate that it is not an artifact of specific datasets or architectures, we conducted extensive additional experiments on 8 diverse regression benchmarks from the UCI Machine Learning Repository (Asuncion et al., 2007). We utilized a multi-layer FT-Transformer (Gorishniy et al., 2021) as the backbone to represent modern, high-capacity attention-based regression models.

Figure 7 illustrates the training (left) and validation (right) trajectories for both MSE and PCC. The curves marked with circles represent PCC metric, while the curves without circle represent MSE.

Figure 7: Training and validation curves for multi-layer FT-Transformer on 8 UCI regression datasets. **Left:** Training set metrics. **Right:** Validation set metrics. The curves with **circles** indicate PCC (maximizing shape matching), while curves **without circles** indicate MSE (minimizing magnitude error). In all instances, PCC hits a ceiling (plateau) significantly earlier than MSE, which continues to descend, highlighting the optimization gap.

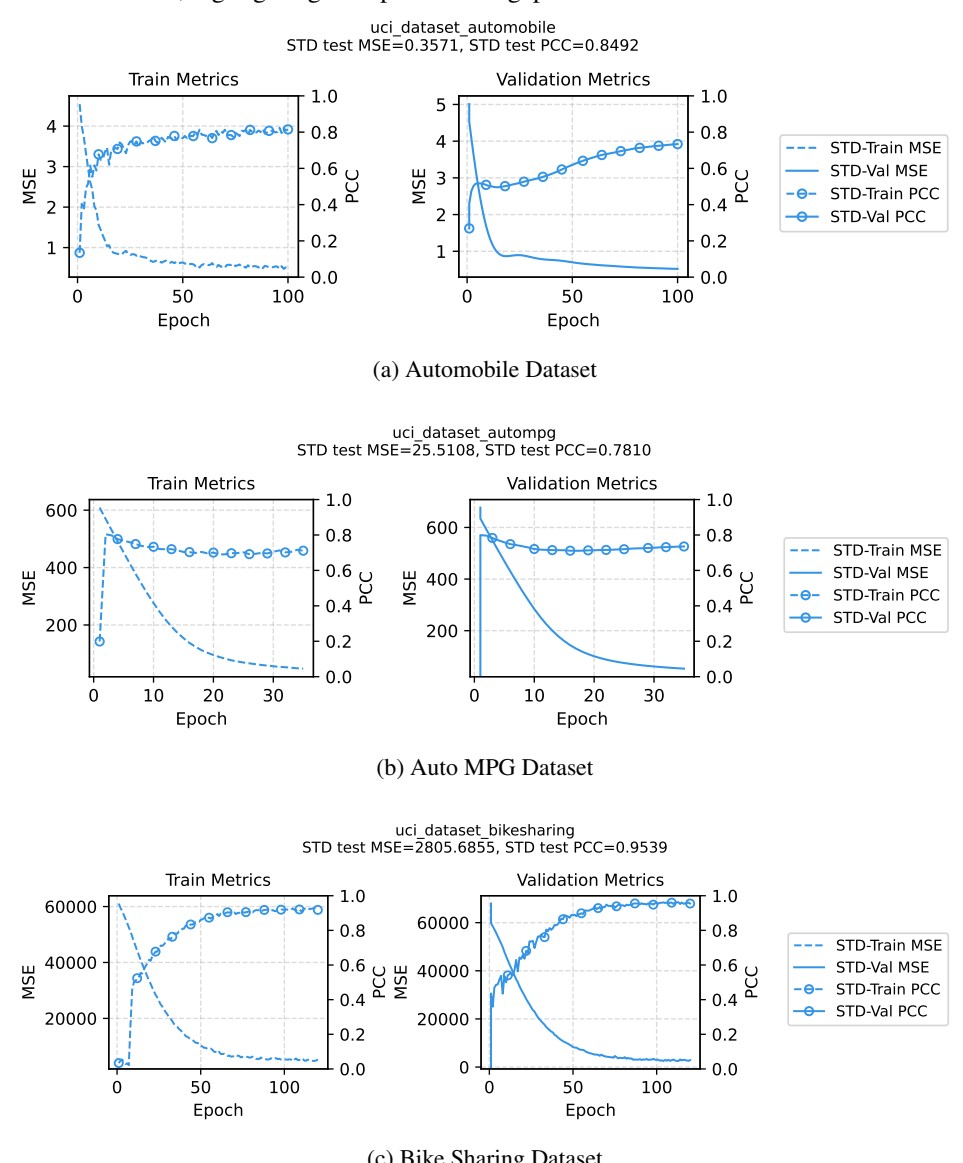

(a) Automobile Dataset

(b) Auto MPG Dataset

(c) Bike Sharing Dataset

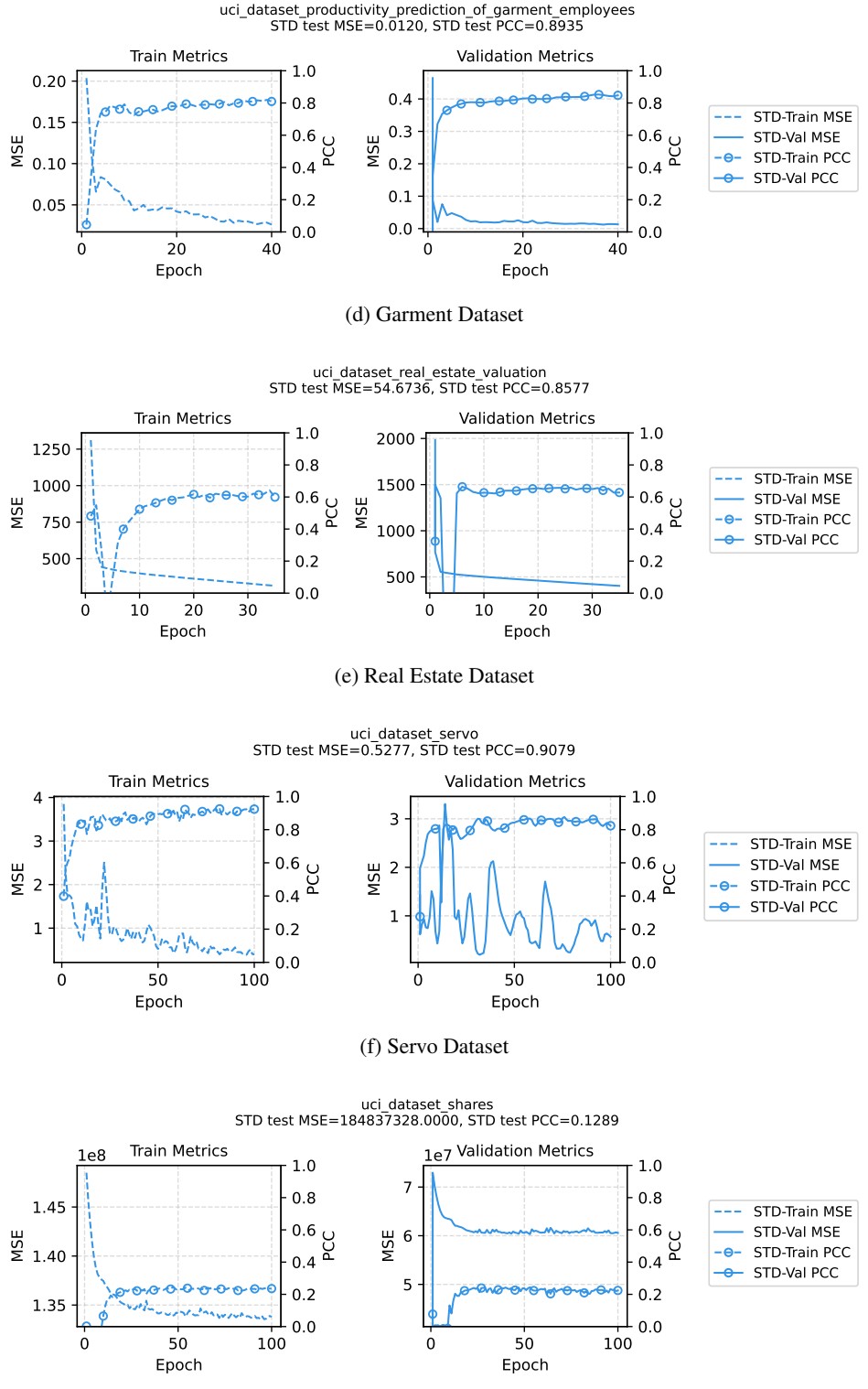

(d) Garment Dataset

(e) Real Estate Dataset

(f) Servo Dataset

(g) Shares Dataset

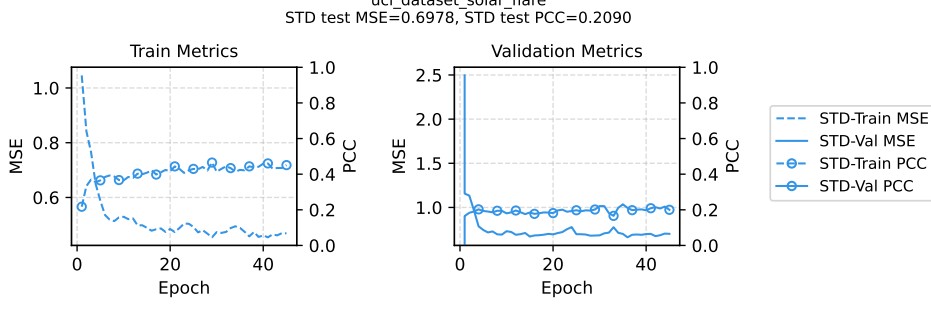

(h) Solar Flare Dataset

**Observations and Analysis:**

- **Decoupling of MSE and PCC:** Across all 8 datasets, we observe a distinct decoupling of the two metrics. In the early stages of training, both MSE and PCC improve. However, a distinct inflection point: the "PCC Plateau" consistently appears where the correlation gain flattens.

- **Persistent MSE Reduction:** Crucially, after the PCC plateaus, the MSE continues to decrease significantly. This confirms the optimization conflict identified in our theoretical analysis (Remark 2.3): the gradient dynamics of MSE minimization (specifically standard deviation matching) actively suppress the correlation gradient magnitude. The model continues to learn magnitude information (lowering MSE) but loses the ability to optimize the shape/ordering of predictions (PCC).

- **Universality:** Despite varying noise levels and feature dimensions across these diverse real-world datasets, the optimization pattern in correlation learning remains identical. This provides robust empirical evidence that the PCC plateau is a fundamental failure mode in joint MSE-PCC training with attention-based models, which calls for a specific architectural improvement as we proposed in Section 3.

## G  DETAILED EXPERIMENTAL SETTINGS

### G.1  SYNTHETIC EXPERIMENT DETAILS

#### G.1.1  DETAILED DATA GENERATION PROCESS

We provide a detailed description of the synthetic dataset generation process (DGP). The DGP is parameterized by the embedding dimension $D$, the number of elements per sample $K$, the signal contrast $\eta$, the noise level $\nu$, the ground truth extrapolation factor $\gamma^* \geq 1$, the cross-sample variation scale $\sigma_B$, and a noise floor $\sigma_{\text{floor}}$.

**1. Defining Ground Truth Directions.**   We first establish the direction of the signal and the direction of the noise.

1. Sample the ground truth regression vector (signal direction) $\mathbf{w}^* \sim \mathcal{N}(0, \mathbf{I}_D)$. Normalize $\|\mathbf{w}^*\|_2 = 1$.
2. Sample a noise direction vector $\mathbf{w}^\perp$. We ensure orthogonality $((\mathbf{w}^*)^\top \mathbf{w}^\perp = 0)$ using the Gram-Schmidt process and normalize $\|\mathbf{w}^\perp\|_2 = 1$.

**2. Generating Sample Centers.**   For each sample $s \in \{1, \ldots, S\}$, we sample the sample center (mean embedding) from an isotropic Gaussian distribution, controlling the cross-sample variation:

$$\boldsymbol{\mu}_s \sim \mathcal{N}(0, \sigma_B^2 \mathbf{I}_D). \tag{96}$$

**3. Generating Elements.**   We generate $K$ elements for each sample $s$, distinguishing between $K - 1$ background samples and one key sample.

**Background Elements** ($i = 1, \ldots, K - 1$): These elements are generated by adding small isotropic noise (noise floor) to the sample center:

$$\mathbf{h}_{si} = \boldsymbol{\mu}_s + \boldsymbol{\epsilon}_{si}, \quad \boldsymbol{\epsilon}_{si} \sim \mathcal{N}(0, \sigma_{\text{floor}}^2 \mathbf{I}_D). \tag{97}$$

**Key Element** ($i = K$): The key element is generated by explicitly injecting signal contrast along $\mathbf{w}^*$ and uninformative noise along $\mathbf{w}^\perp$:

$$\mathbf{h}_{sK} = \boldsymbol{\mu}_s + \underbrace{\eta \cdot \mathbf{w}^*}_{\text{Signal Contrast}} + \underbrace{\nu \cdot \mathbf{w}^\perp}_{\text{Uninformative Noise}} + \boldsymbol{\epsilon}_{sK}. \tag{98}$$

**4. Defining the Target Variable.** The target variable is defined based on an optimal representation $\mathbf{v}_s^*$ that extrapolates the *signal component* by the factor $\gamma^*$:

$$\mathbf{v}_s^* = \boldsymbol{\mu}_s + \gamma^* \cdot (\eta \cdot \mathbf{w}^*). \tag{99}$$

The target label $y_s$ is generated by applying the ground truth regression vector $\mathbf{w}^*$ to this optimal representation, with added label noise $\epsilon_s' \sim \mathcal{N}(0, \sigma_{\text{label}}^2)$:

$$y_s = (\mathbf{w}^*)^\top \mathbf{v}_s^* + \epsilon_s'. \tag{100}$$

### G.1.2 EXPERIMENTAL SETUP AND HYPERPARAMETERS

**Model Architecture.** We use a simple gating attention architecture. The element embeddings $\mathbf{h}_{si}$ are generated directly by the DGP (no pre-trained encoder). The attention logits are computed via a linear layer parameterized by $\mathbf{w}_{\text{attn}} \in \mathbb{R}^{D \times 1}$: $z_{si} = \mathbf{h}_{si}^\top \mathbf{w}_{\text{attn}}$. The aggregated sample-level representation $\mathbf{v}_s$ is computed using $\mathbf{v}_s = \sum_{i=1}^{n_s} \text{Softmax}(\{z_{si}\}_{i=1}^{n_s}) \mathbf{h}_{si}$. The final prediction is computed via a regression head $\mathbf{W}_{\text{reg}} \in \mathbb{R}^{D \times 1}$ and bias $b$: $\hat{y}_s = \mathbf{v}_s^\top \mathbf{w} + b$.

**Optimization.** We train the models using the Adam optimizer (Kingma, 2014) with standard parameters ($\beta_1 = 0.9, \beta_2 = 0.999, \epsilon = 10^{-8}$). The objective function is the joint loss $\mathcal{L}_{\text{total}} = \mathcal{L}_{\text{MSE}} + \lambda_{\text{PCC}} \mathcal{L}_{\text{PCC}}$ (or $\tilde{\mathcal{L}}_{\text{PCC}}$ in Equation (11) for DNPL).

**Hyperparameters.** The default hyperparameters used in the synthetic experiments are summarized in Table 4.

Table 4: Hyperparameters for Synthetic Experiments

| Parameter | Value |
|---|---|
| *DGP Settings* | |
| $D$ (Embedding dimension) | 16 |
| $K$ (Elements per sample) | 10 |
| $N_{\text{train}}$ / $N_{\text{val}}$ | 2000 / 300 |
| $\sigma_B$ | 1.0 |
| $\sigma_{\text{floor}}$ / $\sigma_{\text{label}}$ | 0.01 / 0.01 |
| *Training Settings* | |
| Optimizer | Adam |
| Learning Rate | 0.001-0.01 |
| Epochs | 1000 |
| $\lambda_{\text{PCC}}$ (PCC loss weight) | 0.3-0.8 |
| *ECA Settings* | |
| DATS $T_{\min}$ / $\beta$ | 0-0.8 / 0.5-3 |
| SRA $\lambda_\gamma$ / $\gamma_{\max}$ | 0.001 / 2 |

### G.2 UCI ML REPOSITORY DATASETS

For each dataset, We report PCC, MSE and MAE between predictions $\hat{y}_i$ and ground-truth targets $y_i$ over all samples in test set is

$$\text{PCC}(\hat{y}, y) = \frac{\text{cov}(\hat{y}, y)}{\sigma(\hat{y})\, \sigma(y)}.$$

Mean squared error (MSE) and mean absolute error (MAE) are computed sample-wise and averaged over all gene–window pairs:

$$\mathrm{MSE}\,(\hat{y}, y) = \frac{1}{N}\sum_{i=1}^{N}(\hat{y}_i - y_i)^2, \qquad \mathrm{MAE}\,(\hat{y}, y) = \frac{1}{N}\sum_{i=1}^{N}|\hat{y}_i - y_i|.$$

**1. Appliance Dataset**   The Appliances Energy Prediction dataset from the UCI Machine Learning Repository contains experimental data collected over 4.5 months in a low-energy building. It includes 19,735 instances with 28 real-valued features, including indoor temperature, humidity, and weather conditions. These attributes are recorded at 10-minute intervals. The goal is to predict appliance energy usage as an integer value in watt-hours (Wh).

We replace the attention layer in the FT-Transformer (Gorishniy et al., 2021) with our ECA module and keep the remaining components unchanged. We use the following hyperparameters during training: "batch_size=128", "dropout_rate=0.1", "embedding_dim=256" and "num_heads=8" for both the FT-Transformer baseline and the FT-Transformer+ECA model. We use MSE and MAE to evaluate magnitude matching and PCC to evaluate correlation matching.

**2. Online News Dataset**   The Online News Popularity dataset contains 39,797 news articles from Mashable, each described by 58 numeric features (with the URL and timedelta features excluded). Its target is the article's number of social-media shares (popularity) and we applied the log transformation on the target to reduce the large range of data.

We replace the attention layer in the FT-Transformer (Gorishniy et al., 2021) with our ECA module and keep the remaining components unchanged. We use the following hyperparameters during training: "batch_size=128", "dropout_rate=0.1", "embedding_dim=256" and "num_heads=8" for both the FT-Transformer baseline and the FT-Transformer+ECA model. We use MSE and MAE to evaluate magnitude matching and PCC to evaluate correlation matching.

**3. Superconductivity Dataset**   The UCI Superconductivity dataset contains 21,263 superconductors with 81 real-valued features, where the target is the critical temperature.

We replace the attention layer in the FT-Transformer (Gorishniy et al., 2021) with our ECA module and keep the remaining components unchanged. We use the following hyperparameters during training: "batch_size=128", "dropout_rate=0.1", "embedding_dim=256" and "num_heads=8" for both the FT-Transformer baseline and the FT-Transformer+ECA model. We use MSE and MAE to evaluate magnitude matching and PCC to evaluate correlation matching.

## G.3   SPATIAL TRANSCRIPTOMIC DATASET

**Data Processing.**   We follow the data processing as the EGN baseline (Yang et al., 2023). The 10xProteomic dataset contains $\sim$32,032 image–patch/expression pairs curated from 5 whole-slide images (10x Genomics, 2025; Yang et al., 2023). Prediction targets are restricted to the 250 genes with the largest mean expression across the dataset (computed over all patches). For each target gene, we apply a log transform to expression values and then perform per-gene min–max normalization to map values into $[0, 1]$. We use the dataset-provided image patches without modifying tiling or slide-level grouping.

**Evaluation Metric.**   We report PCC aggregated across genes and sample-wise error metrics. For each gene $g$, PCC between predictions $\hat{y}_g$ and ground truths $y_g$ over all evaluation windows is

$$\mathrm{PCC}(g) = \frac{\mathrm{cov}(\hat{y}_g, y_g)}{\sigma(\hat{y}_g)\,\sigma(y_g)}.$$

We summarize $\{\mathrm{PCC}(g)\}$ across all target genes using: PCC@F (25th percentile), PCC@S (median), and PCC@M (mean). Higher is better.

Mean squared error (MSE) and mean absolute error (MAE) are computed sample-wise and averaged over all gene–window pairs:

$$\mathrm{MSE} = \frac{1}{N}\sum_{i=1}^{N}(\hat{y}_i - y_i)^2, \qquad \mathrm{MAE} = \frac{1}{N}\sum_{i=1}^{N}|\hat{y}_i - y_i|.$$

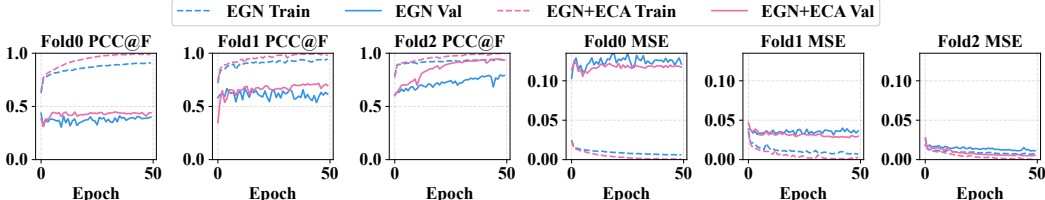

Figure 8: Per-fold PCC@F comparison of EGN and EGN+ECA on 10xProteomic dataset.

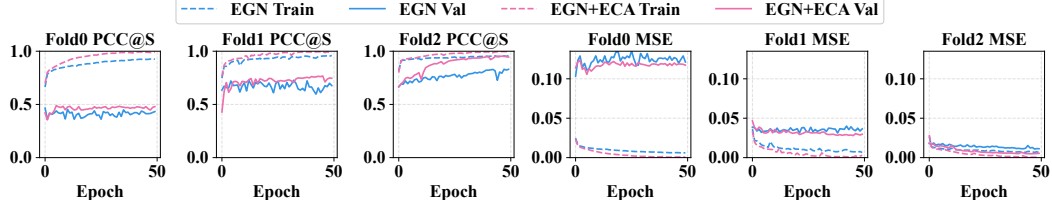

Figure 9: Per-fold PCC@S comparison of EGN and EGN+ECA on 10xProteomic dataset.

**Implementation.** The EGN and ECA improved version are implemented in PyTorch. The visual backbone is a ViT with patch size 32, embedding dimension 1024, MLP dimension 4096, 16 attention heads, and depth 8. We interleave the proposed Exemplar Block (EB) every two ViT blocks. Each EB uses 16 heads with head dimension 64, retrieving the $k=9$ nearest exemplars. Results are reported on a three-fold cross-validation with the same experimental setting in EGN. We train for 50 epochs with a batch size of 32. Experiments run on $2\times$ A6000 GPUs. We apply the same pre-processing and gene-selection protocol throughout and keep dataset-provided tiles and slide/patient groupings intact for fair comparison.

**Extra Results.** Figure 8 and Figure 9 provide extra visualization results on PCC@S and PCC@F metrics for each fold.

## G.4 MULTIMODAL SENTIMENT ANALYSIS (MSA) DATASET

**Data Processing** We conduct multimodal sentiment analysis (MSA) experiments on the CMU-MOSI dataset (Zadeh et al., 2016), a standard benchmark comprising 2,199 opinion-centric video segments. Each segment pairs a short monologue clip with synchronized *language*, *acoustic*, and *visual* modalities and is annotated with a continuous sentiment intensity score on a seven-point scale from $-3$ (strongly negative) to $+3$ (strongly positive). Following the widely adopted split, the data are partitioned into 1,284 / 229 / 686 segments for train/validation/test. In line with community practice and THUIAR's public releases (Yu et al., 2020; 2021), we use pre-extracted, unaligned features where acoustic streams are sampled at 12.5 Hz and visual streams at 15 Hz; text features are aligned to the same timeline.

**Task formulation and metrics.** We evaluate models in the regression setting (predicting the continuous sentiment score) while also reporting the standard classification metric used in prior MSA work. Concretely, we report: (i) MAE: mean absolute error between predicted and ground-truth sentiment; (ii) PCC: Pearson's correlation coefficient between predictions and ground truth, measuring rank/linear consistency; and (iii) F1: the binary F1 score computed under the standard MOSI protocol from prior work and THUIAR's toolkit (Yu et al., 2020; 2021). This suite jointly captures absolute deviation (MAE), correlation structure (PCC), and discrete decision quality (F1) and is consistent with the prevailing MOSI evaluation practice.

**Baselines.** To situate our method among representative approaches, we compare against widely cited MSA baselines that span tensor fusion, low-rank factorization, recurrent and deep feed-forward fusion, graph-based memory models, and cross-modal transformers. Below we list each baseline with its canonical citation (the superscripts indicate the source of numbers when we report them in

tables: [†] from THUIAR's GitHub/toolkit, [*] from the original paper, and [**] reproduced by us with released code/hyperparameters):

- **TFN**[†] (Zadeh et al., 2017): Tensor Fusion Network introducing explicit multimodal tensor interactions.
- **LMF**[*] (Liu et al., 2018): Low-Rank Multimodal Fusion reducing TFN's cubic complexity via factorization.
- **EF-LSTM**[†] (Williams et al., 2018b): Early-fusion recurrent model that concatenates modalities prior to sequence modeling.
- **LF-DNN**[†] (Williams et al., 2018a): Late-fusion deep network combining modality-specific predictors at the decision level.
- **Graph-MFN**[†] (Zadeh et al., 2018): Graph-structured Memory Fusion Network for cross-view temporal reasoning.
- **MulT**[*] (Tsai et al., 2019): Cross-modal Transformer that performs directional attention across modalities.
- **MISA**[†] (Hazarika et al., 2020): Modality-Invariant/Specific representations with contrastive objectives to disentangle factors.
- **ICCN**[*] (Sun et al., 2020): Cross-modal correlation networks leveraging canonical correlation constraints.
- **DLF**[**] (Wang et al., 2025): A recent strong multimodal baseline emphasizing deep latent fusion.
- **ALMT**[**] (Zhang et al., 2023): Attention-based latent model trained with MSE; we use ALMT as the host architecture for our ECA augmentation.

**Implementation.** Unless otherwise noted, we follow the standard MOSI preprocessing and the train/validation/test partition described above. For baselines, we adhere to the configuration choices recommended by the original authors or by THUIAR's toolkit (Yu et al., 2020; 2021). Our primary report includes F1, PCC, and MAE on the held-out test set, with validation used solely for early stopping and hyperparameter selection. The model training is with 2 A6000 GPUs.

We adopt ALMT as the base architecture and keep the optimizer and data pipeline fixed across all runs to ensure a clean ablation. We train for 100 epochs with a batch size of 64, using AdamW with lr=1e-4 and batching and loading use num_workers=8. Other hyperparameters are shown in Table 5.

ALMT, in its original form, is trained with MSE loss only. To probe correlation-aware training and our correlation-aware aggregation, we evaluate two controlled settings (all other hyperparameters are held constant):

1. **ALMT + PCC loss (no architectural change).** We augment the training objective with a differentiable PCC term, yielding a weighted sum

$$\mathcal{L} \; = \; \lambda_{\text{MSE}} \cdot \text{MSE} \; + \; \lambda_{\text{PCC}} \cdot \mathcal{L}_{\text{PCC}},$$

where $\lambda_{\text{MSE}}$=1.0 and $\lambda_{\text{PCC}}$=1.0. This setting isolates the effect of explicitly encouraging high Pearson correlation during training while leaving the model architecture unchanged.

2. **ALMT + PCC loss + ECA (ours).** In addition to the above loss, we adopt our proposed ECA method in the vision stream by modifying the ViT-style token aggregation step.

**Extra Results Analysis.** By comparing Setting 1 to the MSE-only ALMT, we measure the gain from correlation-aware training alone. Comparing Setting 2 to Setting 1 isolates the benefit of ECA's vision-token aggregation. As summarized in Table 2, adding PCC improves correlation-oriented metrics without harming classification quality, and coupling PCC with our ECA aggregation *further* improves all three metrics (F1, PCC, MAE). These results indicate that ECA enhances the multimodal pipeline beyond loss-level changes, demonstrating tangible benefits for *multimodality* on MOSI.

Table 5: Configuration for MOSI experiments (ALMT backbone).

| Param | Value |
|---|---|
| Optimizer / weight decay / lr | AdamW / $1 \times 10^{-4}$ / $1 \times 10^{-4}$ |
| Batch size / epochs | 64 / 100 |
| Text / audio / vision dims | 768 / 5 / 20 |
| Projection dim (all modalities) | 128 |
| Token count / token dim | 8 / 128 |
| Seq. lengths (l/a/v) | 50 / 375 / 500 |
| Projection transformer (depth/heads/MLP) | 1 / 8 / 128 |
| Text encoder (heads/MLP) | 8 / 128 |
| Hyper-layer (depth/heads/head-dim/dropout) | 3 / 8 / 16 / 0 |
| Loss weights (MSE / PCC) | 1.0 / 1.0 |
| ECA module | ✓ in Setting 2 |

# H    BATCH SIZE EFFECT ON PCC PLATEAU

A potential concern regarding the joint optimization of MSE and PCC is whether the batch size $S$ influences the optimization dynamics, specifically the onset of the PCC plateau. Theoretically, as discussed in Section 2.3 and the Equation (6), both the MSE and PCC gradient magnitudes scale proportionally with $1/S$. Consequently, the ratio between the two gradients that determines the effective optimization direction is invariant to the batch size. This suggests that the conflict causing the plateau is fundamental to the gradient properties (specifically the $1/\sigma_{\hat{y}}$ attenuation) and the homogeneous PCC bound, rather than batch statistics.

To empirically validate this independence, we conducted an ablation study using the synthetic dataset described in Section 5.1, training the standard attention regression model with varying batch sizes $S \in \{32, 64, 128\}$.

Figure 10: MSE and PCC curve under various batch size settings.

Synthetic Dataset Eta=0.1, Gamma*=1.0, Lambda_PCC=1.0, Batch_size=32

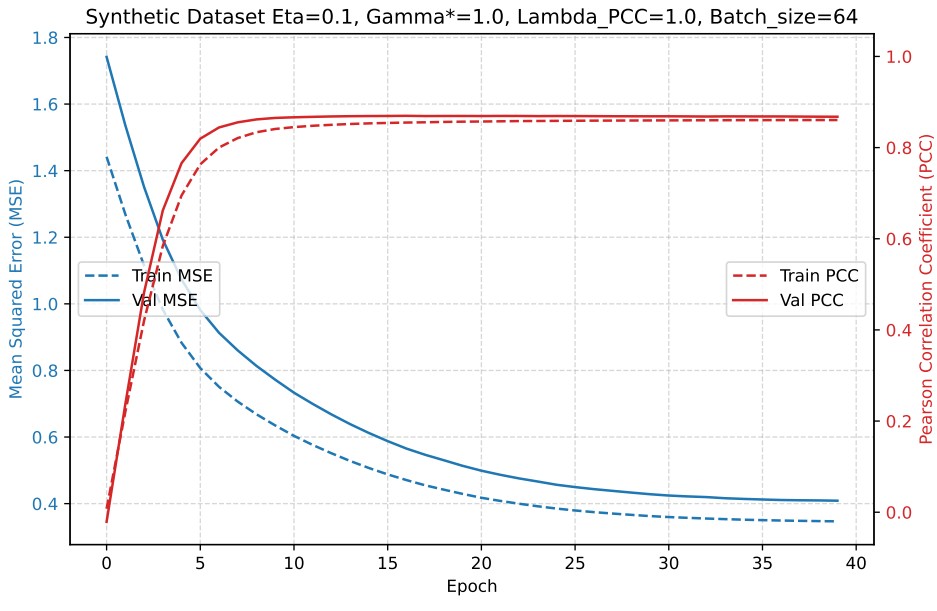

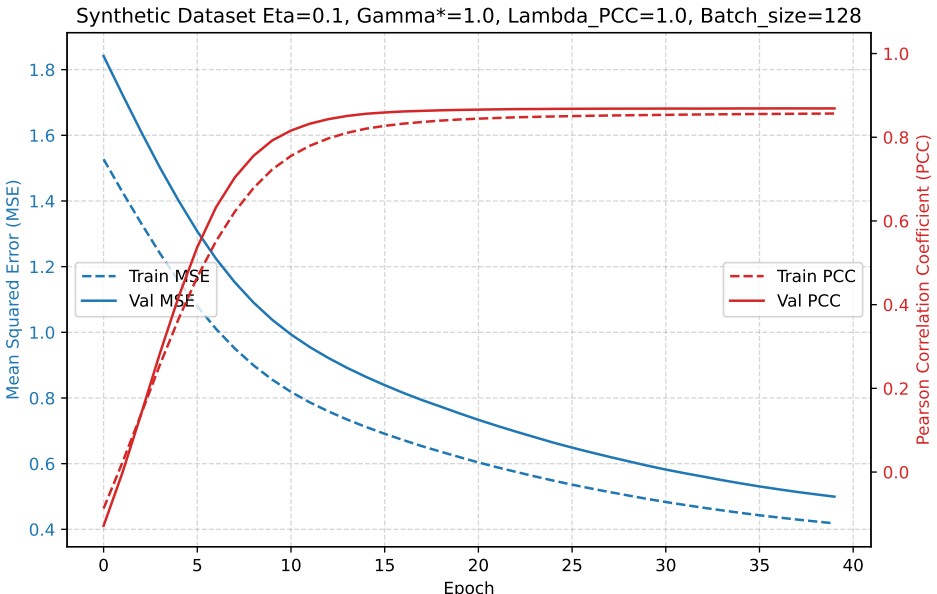

Appendix H presents the learning curves for these settings. The results align with our theoretical analysis:

- **Consistent Plateau Behavior:** Across all batch sizes, we observe the characteristic decoupling of the metrics. The PCC curves (red) flatten and plateau significantly earlier than the MSE curves (blue), which continue to decrease steadily throughout training.

- **Generalization Gap:** Both training (dashed) and validation (solid) curves exhibit similar trends, indicating that this phenomenon is an optimization issue rather than a generalization issue.

These results confirm that increasing or decreasing the batch size does not resolve the PCC plateau, further validating the necessity of the architectural interventions proposed in our ECA to explicitly counteract gradient attenuation.

## I  DISCLOSURE OF LLM USAGE

A large language model (LLM) was used only for language refinement (grammar, phrasing, and clarity) in the main paper and the appendix. The LLM did not contribute to the research ideas, mathematical derivations, theorems, proofs, or experimental design. All technical content was independently produced and reviewed by the authors.

