# OpenReview forum: "Breaking the Correlation Plateau: On the Optimization and Capacity Limits of Attention-Based Regressors"
_ICLR.cc/2026/Conference — ICLR 2026 Poster_

### Official Review · Reviewer_W6BH · 2025-10-28

**Soundness:** 2
**Presentation:** 3
**Contribution:** 2
**Rating:** 4
**Confidence:** 2

**Summary:**

This paper investigates the "PCC plateau" phenomenon in attention-based regression models trained with a joint MSE and PCC loss, where PCC stops improving early even as MSE decreases. The authors provide a theoretical analysis identifying two main causes: 1) an optimization conflict where minimizing MSE paradoxically suppresses the PCC gradient, exacerbated by softmax attention, especially with homogeneous input elements within a sample; and 2) a model capacity limit where convex aggregators like softmax attention restrict PCC improvements based on the convex hull radius of the input embeddings. To overcome these limitations, the paper proposes Extrapolative Correlation Attention, incorporating a dispersion-normalized PCC loss, dispersion-aware temperature softmax, and scaled residual aggregation to stabilize PCC optimization, adapt to homogeneity, and allow extrapolation beyond the convex hull. Experiments across various benchmarks demonstrate that ECA successfully breaks the PCC plateau and improves correlation without degrading MSE performance.

**Strengths:**

- The paper provides a thorough theoretical investigation into the PCC plateau phenomenon from two complementary perspectives: optimization dynamics and model capacity. This analysis successfully identifies key bottlenecks inherent in standard softmax attention for correlation learning, specifically the gradient attenuation conflict with MSE and the limitations imposed by convex aggregation.
- The proposed Extrapolative Correlation Attention (ECA) method directly addresses the theoretically identified bottlenecks. Each component of ECA is well-motivated by the preceding analysis, offering targeted solutions to the optimization and capacity limitations.

**Weaknesses:**

- While the theoretical analysis provides valuable insights, certain aspects could benefit from further development or more rigorous treatment. Specific points regarding the theoretical sufficiency will be elaborated upon in the questions section.

**Questions:**

- While Corollary 2.1 provides a bound on the PCC gradient magnitude, it's not immediately clear if this bound alone sufficiently explains the relative slowdown of PCC compared to MSE during optimization. Could the authors provide a more direct comparison, perhaps by analyzing the relative magnitudes or scaling of the PCC gradient versus the MSE gradient components ?
- For a more comprehensive understanding of the optimization dynamics, would analyzing the system using gradient flow provide further insights or more robust theoretical guarantees regarding the plateau phenomenon?
- The central claim is that addressing the identified gradient attenuation and capacity bottlenecks improves PCC performance. Could the authors elaborate further on the core intuition for why mitigating these specific issues leads to better correlation learning?

---

> ### Author Response · Authors · 2025-11-21
>
> We thank you for your careful reading of our manuscript and for your insightful remarks. We are pleased that you regard the theoretical analysis as “valuable”, the proposed method as “well-motivated,” and we “successfully identify key bottlenecks”. Below, we provide detailed responses to each of your questions.

---

> ### Author Response · Authors · 2025-11-21
> **Our clarification regarding the reviewer's concerns and questions.**
>
> ### **W1: Gradient Comparison Between PCC and MSE**
> We thank the reviewer for this suggestion and have added an explicit comparison between the PCC and MSE gradients w.r.t. the attention logits. We show that using Lem. 2.1 (Softmax Jacobian), the MSE loss has gradient:
> $$
> \frac{\partial L_{\mathrm{MSE}}}{\partial z_{si}} = \frac{2}{S}(\hat y_s - y_s)\alpha_{si}\mathbf{w}^\top(\mathbf{h}_{si}-\mathbf{v}_s).
> $$
>
> And according to the Thm. 2.1, the PCC has gradient:
> $$
> \frac{\partial \rho}{\partial z_{si}} = \frac{1}{S\sigma_{\hat y}}\left(\frac{a_s}{\sigma_y} - \rho\frac{b_s}{\sigma_{\hat y}}\right)\alpha_{si}\mathbf w^\top(\mathbf h_{si}-\mathbf v_s)
> $$
>
> Thus, up to a common factor $\alpha_{si}\mathbf w^\top(\mathbf h_{si}-\mathbf v_s)$, the difference between the MSE and PCC gradients is entirely the following global scalar terms:
> $
> g_s^{\mathrm{MSE}} := 2(\hat y_s - y_s),
> \quad
> g_s^{\mathrm{PCC}} := \frac{1}{\sigma_{\hat y}}
> \left(\frac{a_s}{\sigma_y}-\rho\frac{b_s}{\sigma_{\hat y}}\right)\.
> $
>
> We define the ratio between the root-mean-square (RMS) of the two scalar terms over the dataset samples as:
> $$
> r_{\text{diff}}
> := \frac{\sqrt{\mathbb{E}_s[(g_s^{\mathrm{PCC}})^2]}}
>         {\sqrt{\mathbb{E}_s[(g_s^{\mathrm{MSE}})^2]}}\.
> $$
>
> We can theoretically show that this ratio is bounded by:
> $$
> r_{diff} = \frac{\sqrt{E_s[(g_s^{PCC})^2]}}{\sqrt{E_s[(g_s^{MSE})^2]}} = \frac{\sqrt{1-\rho^2}}{2\sigma_{\hat y}\sqrt{MSE}} \le \frac{1}{2\sqrt{\sigma_y}\sigma_{\hat y}^{3/2}}
> $$
>
> We have added this theoretical analysis of the ratio between the root-mean-square MSE and PCC gradients as **Corollary 2.1** in the main text of the revision version and added the complete proof into the **Appendix C.2**. We also validate this $r_{diff}$ bound on synthetic dataset in Figure 3 (revision version).The RMS ratio of PCC vs. MSE gradients (blue) is strictly constrained by the theoretical upper bound (red dashed). The increase in prediction dispersion $\sigma_{\hat{y}}$ (green) during training drives the rapid attenuation of the PCC gradient signal relative to the MSE gradient, which confirms the relative decay ratio of at least order $\mathcal{O}(1/\sigma_{\hat y}^{3/2})$.
>
> ### **Q2: On Gradient-Flow Analysis of the Plateau Phenomenon**
> Thank you for bringing up the insightful gradient-flow perspective. Our result can indeed be interpreted at the level of gradient flow by writing the joint objective as
> $$
> L = MSE + \lambda (1 - \rho)
> $$
> and the corresponding gradient-flow dynamics are
> $$
> \dot z = -\nabla_z L = -\nabla_z MSE + \lambda \nabla_z \rho
> $$
> Our analysis already characterizes $ \|\nabla_z MSE\| $ (rebuttal for Question 1 and Lemma 2.3 in revision pdf) and $ \|\nabla_z \rho\| $. In the small-step-size limit, gradient descent converges to this ODE, and the same relative scaling (PCC/MSE ratio decays at least order $\mathcal{O}(1/\sigma_{\hat y}^{3/2})$) implies that the flow is dominated by the MSE term as $\sigma_{\hat y}$ gets large, leading to an early plateau in $\rho$. We will clarify this connection in the revision.
>
> Deriving global convergence guarantees for this non-convex, data-dependent system would require strong additional assumptions. While a full gradient-flow perspective is beyond the scope of this work, we believe our analysis provides sufficient theoretical grounding for the PCC plateau phenomenon, and we agree that a more detailed gradient-flow analysis is an interesting direction for future work.

---

> ### Author Response · Authors · 2025-11-21
>
> ### **Q3: Clarifying Intuition Behind Bottleneck Mitigation**
> We thank the reviewer for asking for more intuition on the identified bottlenecks. The core intuition is that achieving high correlation requires the model to detect and represent subtle contrasts across samples that determine the correct relative ordering (PCC) of predictions, especially when in-sample homogeneity is higher than cross-sample homogeneity. Standard attention fails at both. Our analysis shows that:
>
> 1. **Gradient Attenuation**: PCC is scale-invariant, but its gradient is not. The $1/\sigma_{\hat{y}}$ factor makes the PCC gradient signal vanish as calibrating MSE loss generally increases the $\sigma_{\hat{y}}$ by matching ground-truth $\sigma_y$. Our proposed DNPL module explicitly removes this attenuation (via the stop-grad scaling), so the correlation term keeps applying meaningful influence late in training.
>
> 2. **Capacity Bottleneck**: convex attention confines the output to a small region determined by the convex hull of the in-sample embeddings. When in-sample dispersion is low (high homogeneity), the achievable variation across samples is limited, which bottlenecks the maximum PCC. Our proposed SRA (with extrapolation beyond the convex hull) and DATS (adaptive sharpening) enlarge the effective range of the output embedding, giving the model the representational freedom needed to align with the target ranking and thus improve PCC.

---

> ### Author Response · Authors · 2025-11-27
>
> Dear Reviewer,
>
> We sincerely appreciate your thoughtful review. In our rebuttal, we have addressed all of your questions and the concerns you raised. If anything remains unclear, please feel free to let us know and we would be happy to clarify.
>
> When you have a moment, we would be grateful if you could revisit our responses and consider raising the score if you find our clarifications satisfactory. Thank you again for your time.
>
> Best regards,
>
> The Authors

---

### Official Review · Reviewer_pBmc · 2025-10-29

**Soundness:** 3
**Presentation:** 3
**Contribution:** 2
**Rating:** 4
**Confidence:** 4

**Summary:**

This paper studies the *PCC plateau* phenomenon where for attention-based regression models, PCC stops improving early in training even as MSE continues to decrease. One of the contribution of the paper lies in the theoretical characterization of the following conflict: lowering MSE  can paradoxically suppress the PCC gradient, particular when the data to be aggregated is highly homogeneous. Moreover, the authors derive a PCC improvement limit for any convex aggregator (including the softmax attention). Finally, to solve the aforementioned issues, the authors propose several tricks to rescale the PCC loss, adjust attention temperature. Empirical results prove their proposed methods work well in practice.

**Strengths:**

**Novel topics and clear interpretation:** I like the topic studied in this paper since learning PCC is an important problem, and there lacks theoretical understanding of why the PCC plateau happens. Moreover, the decomposition in Proposition 2.1 offers a clean interpretation of how the matching of mean, std, and correlation affect the MSE.

**Clear theoretical analysis** In Section 2, the authors provide several propositions and theorems to argue why the PCC plateau happens by analyzing the gradients. Though the analysis has some weakness (see weakness section for details), these theoretical results provide a clean interpretations of how the homogeneity of the data, batch size, cross-sample deviation affect the overall gradients. Considering this work is the first to address this issue, I appreciate their contributions.

**Strong empirical validation:** In the experiment section, the authors evaluate their proposed methods across four different datasets, and demonstrate sufficient improvement of the increase of PCC when their method has been applied. Moreover, the authors conduct ablation studies to identify the contributions of each component of their proposed algorithm.

**Weaknesses:**

**Missing comparison of gradient between MSE and PCC:** In Section 2.3, the author present gradient of PCC w.r.t. attention logits (see Theorem 2.1), then provide a bound for this gradient in Corollary 2.1, and finally discuss the implications of the bounds. It is clear how each term affect the gradient for PCC. However, it lacks the discussions of the bounds for MSE as well. Since this paper argues *PCC tends to flat earlier while MSE continues to decrease*, it is important to discuss the gradient of MSE, and compare them to support this claim.

**Weak theoretical contribution:** The theoretical contributions of this work is mainly point-wise comparison (section 2.3 and 2.4). For example, in theorem 2.2, the authors state the bounds for $\rho-\rho_0$ for fixed $\tilde R, \sigma_0, w$. However, it lacks the understanding of how the quantities of interest evolve throughout the training, and what is the effect of them cumulatively on the PCC and MSE.

**Limited discussions on the trade-offs:** This paper introduces a novel objective in equation 8 to improve the learning of PCC. However, adding regularization might affect the performance of MSE, training speed, or stability. However, none of this trade-offs has been discussed in the paper.

**Questions:**

**Question 1** Can you please explain how can we see the conclusion *PCC flattens at early epoch while MSE continues to decrease* in Figure 2? I feel the curves for training mse and training pcc seem to overlap.

**Question 2** In theorem 2.2, because PCC is less than 1, if the RHS of the inequality os larger than 2, it provides meaningless bound. Therefore, the bound is only meaningful when the RHS is substantially less than 2. However, it is unclear whether this quantity satisfies this conditions.

**Question 3** In the experimental sections, are the reposted MSEs training loss or test loss? In Table 1 and Table 2, it seems adding this additional regularization, the MSE also decreases. This is a little surprising to me. The reason is that when you add the additional regularization to improve PCC, essentially you're introducing the bias for the MSE loss, and resulting training loss should not be lower than the one without the regularization. Can you please provide some intuitions why this happens?

---

> ### Author Response · Authors · 2025-11-21
>
> We appreciate your careful review and constructive, encouraging comments. We are pleased that you like our topic and found the theoretical study "clear", the experiments "strong". Below, we answer your questions point by point.

---

> ### Author Response · Authors · 2025-11-21
> **Our clarification regarding the reviewer's concerns and questions.**
>
> ### **W1: Gradient Comparison Between PCC and MSE**
> We thank the reviewer for this suggestion and have added an explicit comparison between the PCC and MSE gradients w.r.t. the attention logits. We show that using Lem. 2.1 (Softmax Jacobian), the MSE loss has gradient:
> $$
> \frac{\partial L_{\mathrm{MSE}}}{\partial z_{si}} = \frac{2}{S}(\hat y_s - y_s)\alpha_{si}\mathbf{w}^\top(\mathbf{h}_{si}-\mathbf{v}_s).
> $$
>
> And according to the Thm. 2.1, the PCC has gradient:
> $$
> \frac{\partial \rho}{\partial z_{si}} = \frac{1}{S\sigma_{\hat y}}\left(\frac{a_s}{\sigma_y} - \rho\frac{b_s}{\sigma_{\hat y}}\right)\alpha_{si}\mathbf w^\top(\mathbf h_{si}-\mathbf v_s)
> $$
>
> Thus, up to a common factor $\alpha_{si}\mathbf w^\top(\mathbf h_{si}-\mathbf v_s)$, the difference between the MSE and PCC gradients is entirely the following global scalar terms:
> $
> g_s^{\mathrm{MSE}} := 2(\hat y_s - y_s),
> \quad
> g_s^{\mathrm{PCC}} := \frac{1}{\sigma_{\hat y}}
> \left(\frac{a_s}{\sigma_y}-\rho\frac{b_s}{\sigma_{\hat y}}\right)\.
> $
>
> We define the ratio between the root-mean-square (RMS) of the two scalar terms over the dataset samples as:
> $$
> r_{\text{diff}}
> := \frac{\sqrt{\mathbb{E}_s[(g_s^{\mathrm{PCC}})^2]}}
>         {\sqrt{\mathbb{E}_s[(g_s^{\mathrm{MSE}})^2]}}\.
> $$
>
> We can theoretically show that this ratio is bounded by:
> $$
> r_{diff} = \frac{\sqrt{E_s[(g_s^{PCC})^2]}}{\sqrt{E_s[(g_s^{MSE})^2]}} = \frac{\sqrt{1-\rho^2}}{2\sigma_{\hat y}\sqrt{MSE}} \le \frac{1}{2\sqrt{\sigma_y}\sigma_{\hat y}^{3/2}}
> $$
>
> We have added this theoretical analysis of the ratio between the root-mean-square MSE and PCC gradients as **Corollary 2.1** in the main text of the revision version and added the complete proof into the **Appendix C.2**. We also validate this $r_{diff}$ bound on synthetic dataset in Figure 3 (revision version).The RMS ratio of PCC vs. MSE gradients (blue) is strictly constrained by the theoretical upper bound (red dashed). The increase in prediction dispersion $\sigma_{\hat{y}}$ (green) during training drives the rapid attenuation of the PCC gradient signal relative to the MSE gradient, which confirms the relative decay ratio of at least order $\mathcal{O}(1/\sigma_{\hat y}^{3/2})$.
>
>
> ### **W2: Concern on Static Bounds**
> We thank the reviewer for raising this concern about the theoretical contribution. We clarify that our theoretical analysis goes beyond a single point-wise comparison.
>
> 1. The MSE decomposition in Sec. 2.2 **holds at every training step** and explicitly couples PCC with the mean and variance terms optimized by MSE, so changes in $\rho$ (PCC) and in MSE are linked throughout training.
>
> 2. The gradient formula and bound (Thm. 2.1 & Cor. 2.2 in revision pdf) show that the PCC gradient w.r.t. attention logits satisfies
> $
> \left|\frac{\partial \rho}{\partial z_{si}}\right| \propto \frac{1}{\sigma_\hat{y}}.
> $
> Under the joint MSE $+\ \lambda_{\text{PCC}}(1-\rho)$ loss, $\sigma_\hat{y}$ is empirically driven toward $\sigma_y$, so the factor $1/\sigma_y$ systematically attenuates PCC gradients while the MSE gradient remains substantial (their ratio decays at $\mathcal{O}(1/\sigma_{\hat y}^{3/2})$ according to Corollary 2.1 in revision). This explains why PCC plateaus earlier whereas MSE keeps decreasing cumulatively over training (as seen in our Fig. 2 and Fig.7 in Appendix).
>
> 3. The PCC gain bound for convex attention (Thm. 2.2) is a "time-uniform" capacity bound: for **any iterattion/epoch** of a convex attention model, the possible PCC improvement over mean pooling is controlled by the homogeneity term $\tilde{R}$ and $\sigma_0 / \|\mathbf{w}\|$.
>
> In summary, the above provides a systematic explanation of how the key quantities dynamically change and why PCC training ultimately gets stuck unless we move beyond convex aggregation, which is exactly what ECA does. To strictly validate this dynamic analysis, we added **Figure. 3** (in revision pdf), which plots the theoretical upper bound of the PCC/MSE gradient ratio against the empirical ratio. The empirical trajectory consistently falls inside the theoretical bound, which tightens as training progresses, confirming our theory accurately models the optimization dynamics.

---

> ### Author Response · Authors · 2025-11-21
>
> ### **W3: Concern on Trade-off Analysis**
> We thank the reviewer for pointing out the need to discuss potential trade-offs. We clarify that the Dispersion-Normalized PCC Loss (DNPL, Eq. 8) is not a regularization term. It is a gradient rescaling designed to counteract the $1/\sigma_{\hat y}$ attenuation identified in Thm 2.1. Specifically,
>
> 1. **MSE Trade-off**: The modified PCC term is $\tilde L_{PCC} = \mathrm{StopGrad}(\sigma_y)(1 - \rho)$, which only rescales gradients and leaves all stationary points of the original $MSE + \lambda_{PCC} (1 - \rho)$ objective (and thus the achievable MSE) **unchanged**. The SRA scale $\gamma_s$ is softly regularized toward 1 and clipped, so when extrapolation is not useful the model collapses back to standard convex attention.
>
> 2. **Speed/Stability**: ECA is lightweight (SRA adds a small MLP, DATS/DNPL are element-wise operations) and introduces negligible computational overhead. Empirically, across all benchmarks we reported, ECA consistently improves PCC while matching or slightly improving MSE/MAE, using the exact same optimizer and without observing instability.
>
> Empirically, ECA consistently improves PCC while matching or slightly improving MSE/MAE across all benchmarks we report when using the same optimizer for both ECA and baseline methods and **did not** observe instability. We will add a short paragraph explicitly discussing these trade-offs in the revision.
>
> ### **Q1: Interpretation on PCC Plateau**
> We apologize for the confusion caused by the unified MSE scale in Figure 2. By “PCC plateau” we refer to the slowdown in the improvement of PCC (i.e., the slope). In Figure 2, for dataset A, which has higher homogeneity (blue curves), the PCC curves flatten relatively early in training (around Epoch 140), while the corresponding MSE curves continue to decrease for many more epochs. This difference in slopes, PCC curve flattens early while MSE keeps improving, is the phenomenon we study. We will revise the figure to make this phenomenon more apparent.
>
> Moreover, to better validate and visualize the PCC plateau in various dataset, we trained multi-layer FT-Transformers on **8 additional UCI regression datasets**. As **Fig.9** shown in the revision pdf Appendix, most dataset exhibits PCC flattening earlier than MSE, providing a straightforward demonstration of this pattern under the attention-based regression settings.
>
> ### **Q2: Practical Tightness of the PCC Bound**
>
> We thank the reviewer for this insightful observation. Yes, the bound in Thm. 2.2 is always valid even when the RHS exceeds 2, in this case it becomes trivial. The bound is informative when the RHS is substantially less than 2, which occurs when the within-sample dispersion ($\tilde{R}$) is significantly smaller than the normalized cross-sample dispersion ($\sigma_{0}/|\mathbf w|_2$).
>
> Empirically, this condition is satisfied in our target domains. For example, in the Spatial Transcriptomics dataset, within-sample dispersion is 0.068 while the cross-sample dispersion is 0.164; Also, in the multimodal sentiment analysis (MSA) dataset, the within-sample dispersion is 0.098 while the cross-sample dispersion is 0.170. The key insight of Thm 2.2 is the functional dependency: as homogeneity increases ($\tilde{R} \to 0$), the bound tightens, and the achievable PCC gain approaches zero.
>
> ### **Q3: Clarification on MSE Metrics and Regularization Effects**
> Thank you for the question. As shown in the legends of all experimental result figures, we report both the training (dashed) and validation (solid) curves for MSE and PCC. The tables in the experimental section reported the test MSE/cross-validation results.
>
> The observation that **both MSE and PCC improve simultaneously is natural** under the MSE decomposition (Prop. 2.1): MSE can be decomposed into mean matching, standard deviation matching and weighted correlation $\rho$. If a model can achieve better correlation (higher $\rho$) while maintaining similar mean and standard deviation matching, the total MSE must decrease.
>
> The intuition that adding regularization increases loss generally applies to training loss with fixed hypothesis space. However, ECA is not simply adding regularization, it fundamentally expands model capacity and improves optimization dynamics:
> 1. SRA allows extrapolation beyond the convex hull, **strictly expanding the model's expressivity** compared to standard attention, which is capacity-limited (Thm 2.2);
>
> 2. DNPL/DATS stabilize the optimization dynamics. By improving optimization and expanding capacity, ECA guides the model toward better feature representations that generalize better, thereby improving both test PCC and test MSE.

---

> ### Comment · Reviewer_pBmc · 2025-11-25
>
> Thank you very much for the detailed responses, and they address my concerns and questions. I will increase the score to 6.

---

### Official Review · Reviewer_4Xnu · 2025-10-30

**Soundness:** 3
**Presentation:** 3
**Contribution:** 3
**Rating:** 6
**Confidence:** 4

**Summary:**

The paper studies a pervasive but underexplored failure mode in attention-based regression models trained with a joint Mean Squared Error (MSE) and Pearson Correlation Coefficient (PCC) objective: the “PCC plateau,” where PCC stops improving early in training even though MSE continues to fall. The authors trace this plateau to two coupled bottlenecks. First, an optimization bottleneck: minimizing MSE increases the variance of the model’s predictions, but the PCC gradient scales inversely with that variance, so the PCC term effectively “goes silent” as training progresses. This effect is exacerbated in settings where each sample’s elements are internally homogeneous, because the PCC gradient flowing through attention weights becomes extremely weak. Second, a capacity bottleneck: standard softmax attention is a convex aggregator, so its output is restricted to the convex hull of the input embeddings. The authors prove that in low-dispersion regimes this imposes a hard upper bound on how much PCC can improve beyond naive mean-pooling.

Guided by this analysis, the paper introduces Extrapolative Correlation Attention (ECA), a drop-in alternative to softmax attention built to overcome both bottlenecks. ECA has three components: (i) Scaled Residual Aggregation (SRA), which explicitly extrapolates beyond the convex hull to increase representational contrast; (ii) Dispersion-Aware Temperature Softmax (DATS), which sharpens attention adaptively when a sample is internally homogeneous; and (iii) Dispersion-Normalized PCC Loss (DNPL), which rescales the PCC objective to keep its gradient active even late in training. Across synthetic, tabular, biomedical, and multimodal benchmarks, ECA prevents the PCC plateau: it continues improving correlation throughout training and delivers higher final PCC while matching or improving standard error metrics (MSE/MAE).

**Strengths:**

Overall, the paper is well written. Here are my understanding of the strength part:

1. The paper identifies and analyzes an interesting “PCC plateau” phenomenon, an observed but under-theorized failure mode in attention-based regression models trained with joint MSE+PCC loss. In my perspective, because the phenomenon emerges in multiple architectures and datasets, the problem is likely to generalize and will be of interest to both theory-leaning and applied ML audiences.

2. On the theory side, the paper decomposes why PCC plateaus: an optimization bottleneck, where minimizing MSE inflates prediction variance and in turn suppresses the effective PCC gradient (especially under homogeneous samples), and a capacity bottleneck, where softmax attention is inherently limited because it forms convex combinations and thus cannot escape a tight convex hull when all elements in a sample are very similar. Crucially, the proposed architectural solution — Extrapolative Correlation Attention (ECA) — is directly motivated by those two factors. This feels principled rather than heuristic.

3. The empirical section is broad, well thought out, and aligned with the claims: The authors test on synthetic data with controlled homogeneity to stress-test exactly the regimes where theory predicts the plateau — and show that ECA keeps PCC improving instead of flattening. They evaluate on standard tabular regression benchmarks, showing consistent PCC gains and equal or better MSE/MAE after swapping in ECA, suggesting the method is not domain-specific. They move to challenging applied domains (spatial transcriptomics, multimodal sentiment) where correlation is a core metric and within-sample homogeneity is very strong. They also include ablations of ECA’s components, which helps establish that the gains are actually due to the proposed mechanisms, not just tuning or extra capacity.

**Weaknesses:**

While the paper provides a principled and well-motivated analysis, several aspects could be further clarified or extended:

1. Architectural depth and generality. Most theoretical and synthetic experiments use a one-layer transformer. It remains unclear whether the proposed ECA mechanism yields similar improvements in deeper architectures, where subsequent layers (especially MLPs) could partially overcome the convex-hull limitation. Without results on multi-layer settings, the contribution may appear as a targeted fix for shallow regressors rather than a generally applicable improvement to attention.

2. Effect of normalization layers. The current analysis does not account for the common use of pre- or post-LayerNorm in standard transformer blocks. Layer normalization can re-center and rescale token representations, which might already mitigate homogeneity and alleviate part of the optimization bottleneck. A more systematic investigation of how normalization interacts with the proposed mechanisms would strengthen the theoretical claims.

3. Baseline coverage and fairness of comparison. The baselines focus mainly on vanilla softmax attention with a joint MSE + PCC objective. Missing comparisons include: (1) attention variants that allow negative or signed weights [1, 2], which can also move beyond convex combinations; and (2) feature-wise normalization or whitening methods that implicitly address variance inflation.
Including these would clarify whether ECA’s gains stem from its theoretical design rather than simply greater expressiveness or modified normalization.

References
- [1] Zhang et al., Negative-Aware Attention Framework for Image-Text Matching, CVPR 2022.
- [2] Lv et al., More Expressive Attention with Negative Weights, arXiv:2411.07176, 2024.

**Questions:**

In general, I like the idea of going beyond the convex hull of softmax attention. However, this limitation may not apply to the linear attention, or other variation of the attention that permits negative attention scores as discussed before. I believe further investigations on how ECA interacts with these non-convex attention would be beneficial.

---

> ### Author Response · Authors · 2025-11-21
>
> Thank you for your careful assessment and for the insightful comments. We are encouraged that you found the problem as “interesting”, the method as “principled,” and the paper as “well written”. We now address each of your questions in turn and in detail.

---

> ### Author Response · Authors · 2025-11-21
> **Our clarification regarding the reviewer's concerns and questions.**
>
> ### **W1: Architectural Depth and Generality**
> Thank you for raising this point. We clarify that our contribution is a **targeted fix for any attention-based regressors** (with linear read-out) and also a generally applicable improvement to attention. Our theoretical analysis intentionally focuses on the final attention aggregation with a linear readout layer, but this is not a shallow special case: it exactly matches the structure used at the output of standard multi-layer Transformers. Formally, we allow an arbitrary backbone $f_\theta$ (any number of Transformer blocks with multi-head attention, FFNs, residual connections, and LayerNorm), and only assume that the final prediction has the form $\hat y_s = w^\top v_s + c$, where $v_s$ is produced by a convex attention pooling over the last-layer token features.
>
> A very common [CLS]-based prediction is a direct instance of this setting. The [CLS] token embedding at the last layer $v_s$ is fed to a linear projection, giving prediction $\hat y_s = w^\top v_s + c$. This last-layer [CLS] embedding itself is obtained by convex (softmax) attention over all token embeddings from the previous transformer layer, so the readout is still a convex aggregation of token features, exactly as assumed in our analysis.
>
> Therefore, our work is focusing on a readout-level phenomenon that applies to standard deep Transformers as long as the final pooling remains convex. This applies to both trainable and frozen pre-trained backbones. Notably, contrastive pre-training often yields highly homogeneous embeddings, exacerbating the achievable PCC upper-bound in downstream tasks (Thm. 2.2).
>
> Empirically, our evaluations (Sec. 5) already utilize deep architectures (e.g., multi-layer FT-Transformer, EGN, and ALMT). We observe the PCC plateau in these models, and applying ECA, which modifies the final aggregation, consistently improves PCC. This validates that our contribution is generally applicable and not limited to shallow regressors. We have clarified this backbone-agnostic scope in the **Sec. 2.1** and **Appendix E** in the revision.
>
> ### **W2: Effect of LayerNorm and Feature Whitening**
> We thank the reviewer for highlighting the role of normalization and whitening. We want to clarify:
>
> 1. **LayerNorm**: LayerNorm operates orthogonally to the issues addressed by ECA. LayerNorm standardizes across the feature dimension for each element $\mathbf{h}_{si}$. In contrast, the homogeneity we analyze ($\sigma_s$, Sec 2.1) measures dispersion across the token/element dimension (the sequence axis). LayerNorm does not necessarily increase the in-sample dispersion $\sigma_s$ or the convex hull radius $R_s$, which contributes to the identified limitations (Corollary 1, Theorem 2). The deep architecture baselines (e.g., FT-Transformer) already incorporate LayerNorm, and we still observe the PCC plateau in our newly added PCC and MSE learning curves on 8 UCI datasets (see **Appendix. F** in revision pdf).
>
> 2. **Feature Whitening**: Global feature whitening changes per-dimension scaling but does not, in general, eliminate the large gap between within-sample and cross-sample dispersion that underlies our bounds. Moreover, feature whitening addresses static feature distributions and does not resolve the dynamic optimization conflict identified in Sec 2.3. Finally, some of our UCI dataset experiments already follow standard preprocessing pipelines (including zero-centering) and still exhibit the plateau. We will clarify this point in the revision.

---

> ### Author Response · Authors · 2025-11-21
>
> ### **W3 & Q1: Baseline Coverage and Negative Attention**
> We thank the reviewer for highlighting [1, 2] as related attempts to go beyond vanilla softmax attention. We agree that relaxing the softmax constraints (e.g., negative weights) is an important direction. We want to clarify:
> **(i)** NAAF method in [1] is **still a convex aggregator**. Under our definition, a “convex aggregator” is exactly as follows:
> $$
> v_s = \sum_i \alpha_{si} h_{si},\quad \alpha_{si} \ge 0,\quad \sum_i \alpha_{si} = 1
> $$
> NAAF’s aggregation step for $\hat{v}_i$ is **exactly** of this form:
>
> $$
> \hat v_i = \sum_j w_{ij} v_j,\quad w_{ij} \ge 0,\quad \sum_j w_{ij} = 1
> $$
>
> The negative branch **never** produces a new representation outside the convex hull, it just **subtracts scalar penalties** from the final similarity score.
>
> So our **capacity bound for any convex aggregator** still applies to NAAF.
> The Cog Attention method in [2] essentially breaks the convex hull and it is a highly potential method that can improve the PCC bound under data homogeneity. The experimental comparison is shown in **(iv)**.
>
> **(ii)** Moreover, our goal is different and more specific from the baselines in [1] and [2]: our ECA is not proposed as a general “better attention,” but as a targeted solution to the PCC plateau phenomenon when jointly optimizing PCC and MSE in attention-based regression.
>
> **(iii)** Finally, our proposed framework is intentionally designed as a plug-in, correlation-aware wrapper that can be applied on top of different base attention kernels, including linear and other non-convex variants. Our framework contains 3 plug-in modules, SRA, DATS and DNPL, each tackles a unique bottleneck identified in our theoretical study. So one can easily use their negative-weights attention and still apply our DATS and DNPL on top. In this sense, our framework is complementary: it implements the theoretical insights about PCC plateau and convex aggregation, while negative-weight attentions primarily target general expressiveness and representational collapse.
>
> **(iv)** To empirically verify our ECA's contribution, we conducted new experiments comparing ECA with Cog Attention [2] on the synthetic dataset under different in-sample homogeneity ($\sigma=0.10$ to $\sigma=0.73$).
> | Homogeneity                     |  0.10      |            | 0.24       |            | 0.42       |            | 0.73       |            |
> |---------------------------------|------------|------------|------------|------------|------------|------------|------------|------------|
> |                                 | MSE        | PCC        | MSE        | PCC        | MSE        | PCC        | MSE        | PCC        |
> | Softmax Attention               | 0.4984     | 0.8240     | 0.8503     | 0.8649     | 1.1066     | 0.9217     | 1.3522     | 0.9582     |
> | Cog Attention [2]               | 0.4812     | 0.8226     | 0.5710     | 0.8741     | 0.6411     | 0.9359     | 0.7446     | 0.9759     |
> | Cog Attention [2] + DNPL + DATS | 0.4777     | 0.8353     | 0.6432     | 0.8962    | 0.5984     | **0.9720** | 0.8015     | 0.9833     |
> | **Ours ECA (SRA + DNPL + DATS)**        | **0.4736** | **0.8386** | **0.5030** | **0.9147** | **0.5089** | 0.9648     | **0.6496** | **0.9874** |
>
> Our proposed framework (SRA+DNPL+DATS) shows higher performance that breaks the PCC plateau under high homogeneity ($\sigma$=0.1) while preserving low MSE. As we mentioned in **(iii)**, since our framework is modular, the Cog Attention can be plugged into our framework as a SRA replacement, on the synthetic dataset with $\sigma=0.42$, the Cog Attention + DNPL + DATS achieves the highest PCC which also indicates the benefits from breaking the convex hull during aggregation.

---

> > ### Comment · Reviewer_4Xnu · 2025-11-27
> >
> > Thanks for the clarification. My evaluation of the work remains unchanged

---

### Official Review · Reviewer_y2UY · 2025-11-01

**Soundness:** 1
**Presentation:** 3
**Contribution:** 1
**Rating:** 4
**Confidence:** 3

**Summary:**

This paper studies the training behavior of attention-based regression models jointly optimized with mean squared error (MSE) and Pearson correlation coefficient (PCC) losses. The authors claim that such models commonly exhibit a “PCC plateau,” where correlation stops improving early in training while MSE continues to decrease. They analyze this through a theoretical decomposition linking MSE and PCC, derive a gradient bound suggesting that PCC gradients shrink with low within-sample dispersion and large prediction variance, and prove a convex-aggregation capacity limit. They propose Extrapolative Correlation Attention (ECA), combining (i) Scaled Residual Aggregation, (ii) Dispersion-Aware Temperature Softmax, and (iii) a Dispersion-Normalized PCC loss. Experiments on synthetic, tabular, biomedical, and multimodal sentiment datasets show small but consistent PCC improvements.

**Strengths:**

1. The writing is well structured and easy to follow. The theoretical derivations are transparent and carefully presented.
2. The work targets an interesting question: how attention-based regressors behave when trained with both magnitude and correlation-based losses. This is a topic that deserves rigorous study and of broad interest to the community.
3. The authors validate their method across a diverse and well-chosen set of benchmarks, which strongly supports the generality and effectiveness of their approach.

**Weaknesses:**

1. The theoretical analysis is based on a simplified model: a single attention aggregation layer followed by a linear head. However, the models used in practice are significantly more complex. The paper does not discuss how other architectural elements might interact with or alleviate the identified problems.

2. The experiments only validate the downstream effects rather than the mechanisms proposed by the theory. The paper would be more convincing with additional empirical evidence that isolate the role of each claim.

3. PCC plateau is described as a widely relevant failure mode, but the authors do not provide any citations or external evidence of this claim. The only evidence they introduce in the beginning of the paper is figure 2 based on their own experiments. If this problem is well documented in the literature, the authors should provide citations. If external citations are truly unavailable, the authors must provide overwhelming evidence of the problem's existence themselves, or potentially reframe their contribution to be the identification and formalization of the problem itself.

4. The condition is theorem 2.2 can make the result meaningless. The tightness of the bound is also not discussed (what if the denominator goes to zero?), and it does not explain or provide any intuition about the training dynamics, it is just about the best achievable PCC after training.

**Questions:**

1. Your argument that data homogeneity suppresses the PCC gradient and causes PCC plateau seems to conflate two different effects: homogeneity makes the pooled representation almost insensitive to attention weights, which would weaken gradients for any regression loss (including MSE), but you present it as if this is unique to PCC. At the same time, you also argue that PCC suffers an additional optimization-specific attenuation which would not affect MSE the same way. Can you clarify which of these you believe is actually responsible for the gradient bottleneck?

2. The claim about the gradient bottleneck assumes that the prediction deviation increases early in training, but that quantity could also in principle decrease. Do you still observe a plateau if the initialization has a high output deviation?

3. Why is the role of the batch size S not explored further? Did you try running the experiments with different batch sizes and see if it has an impact on the PCC plateau?

4. Could you comment on the role of the direction of $w$ in theorem 2.2?

---

> ### Author Response · Authors · 2025-11-21
>
> We appreciate your careful review and your detailed, constructive feedback. We are glad that you view the problem as “interesting”, the experiments as “diverse”, and the paper as “well structured” and “easy to follow”. Below, we respond to your questions individually and in detail.

---

> ### Author Response · Authors · 2025-11-21
> **Our clarification regarding the reviewer's concerns and questions.**
>
> ### **W1: Architectural Depth and Generality.**
> Thank you for raising this point. Our theoretical analysis intentionally focuses on the final attention aggregation with a linear readout layer, but this is not a shallow special case: it exactly matches the structure used at the output of standard multi-layer Transformers. Formally, we allow an arbitrary backbone $\(f_\theta\)$​ (any number of Transformer blocks with multi-head attention, FFNs, residual connections, and LayerNorm), and only assume that the final prediction has the form $\hat y_s = w^\top v_s + c$, where $v_s$​ is produced by a **convex attention pooling** over the last-layer token features.
>
> A very common [CLS]-based prediction is a direct instance of this setting. The [CLS] token embedding at the last layer $v_s$​ is fed to a linear projection, giving prediction $\hat y_s = w^\top v_s + c$. This last-layer [CLS] embedding itself is obtained by convex (softmax) attention over all token embeddings from the previous transformer layer, so the readout is still a convex aggregation of token features, exactly as assumed in our analysis.
>
> Therefore, our work is focusing on a **readout-level phenomenon** that applies to standard deep Transformers as long as the final pooling remains convex. This applies to both trainable and frozen pre-trained backbones. Notably, contrastive pre-training often yields highly homogeneous embeddings, exacerbating the achievable PCC upper-bound in downstream tasks (Thm. 2.2).
>
> Empirically, our evaluations (Sec. 5) already utilize deep architectures (e.g., multi-layer FT-Transformer, EGN, and ALMT). We observe the PCC plateau in these models, and applying ECA, where SRA module modifies the final aggregation, consistently improves PCC. This validates that our contribution is generally applicable and not limited to shallow regressors. We have clarified this backbone-agnostic scope in the Sec. 2.1 and Appendix E in the revision.
>
> ### **W2: Response to Mechanism Validation.**
> We agree that directly validating the underlying mechanisms strengthens the paper. We highlight how our experiments isolate both theoretical claims and provide new analysis:
>
> 1. **Optimization Dynamics (Gradient Attenuation)**: Our theory (Corollary 2 in revision version pdf) claims the ratio of PCC/MSE gradients decays as $\mathcal{O}(1/\sigma_{\hat{y}}^{3/2})$, contributing to the plateau. To provide direct empirical evidence beyond downstream effects, we conducted an additional analysis on the synthetic dataset (by manually controlling the "key deviation" parameter $\eta$ (and thus $\tilde{\sigma}$)). We tracked the RMS magnitudes of the PCC and MSE global gradient factors throughout training. The results are added as Figure 3 (and discussion after Corollary 2.1) in the revision which confirms our prediction: as $\sigma_{\hat{y}}$ increases, the ratio of the PCC gradient magnitude relative to the MSE gradient magnitude rapidly decreases. This directly validates the optimization conflict. Furthermore, the ablation study (Table 1) shows that removing DNPL (our mechanism theoretically designed to counteract this attenuation) consistently harms PCC (e.g., -2% on Appliance dataset), validating its necessity.
>
> 2. **Model Capacity (Convex Hull Constraint)**: Our theory (Theorem 2.2) claims convex aggregation limits PCC gain under high homogeneity ($\tilde{R}$). This mechanism is directly validated by the synthetic experiment (Fig. 4). By controlling the homogeneity level ($\tilde{\sigma}$), we observe that as homogeneity increases (lower $\tilde{\sigma}$), the achievable PCC gain of the standard convex attention baseline shrinks significantly, confirming the capacity limit predicted by the bound.
>
> ### **W3: PCC Plateau Evidence**
> We agree that “PCC Plateau” has not been formally studied or named in prior literature. Our work aims to fill this gap. Following the reviewer's suggestion, we will reframe our contribution: we are the first to formally identify and theoretically analyze this specific empirical phenomenon. This plateau is a mathematical inevitability derived in our optimization dynamic and data homogeneity analysis.
>
> We provide substantial empirical evidence that the effect is not an artifact of a single setup:
> 1. Current Experiments: While real-world noise can visually soften the plateau, it remains detectable as an early reduction in slope (e.g., Spatial Transcriptomics, Fig. 6 in revision), whereas our low-noise synthetic data reveals the plateau explicitly (Fig. 5 in revision).
> 2. New Experiments: To provide the "overwhelming evidence" requested, we trained multi-layer FT-Transformers on **8 additional UCI regression datasets**. As Fig.9 shown in the revision pdf Appendix, most datasets exhibit PCC flattening earlier than MSE, confirming this is a widespread pattern in the attention-based regression settings.

---

> ### Author Response · Authors · 2025-11-21
>
> ### **W4: Theorem Condition & Meaning**
> We clarify the interpretation and scope of Theorem 2.2 (PCC Gain Bound).
> 1. **Capacity vs. Dynamics**: The reviewer is correct that Theorem 2.2 addresses model capacity (Sec 2.4), not training dynamics (Sec 2.3). This is intentional. We separated our analysis into **Optimization Dynamics** (Section 2.3, Theorem 2.1) and **Model Capacity** (Section 2.4, Theorem 2.2). Theorem 2.2 addresses the geometric limit at convergence, while Theorem 2.1 (Gradient) addresses the dynamics.
> 2. **The Condition $\tilde R<\sigma_{0}/\|\mathbf w\|_2$**: This condition defines the regime where the bound is informative and non-trivial. It compares within-sample dispersion ($\tilde{R}$) to normalized cross-sample variation ($\sigma_0/\|\mathbf w\|_2$). The condition requires that in-sample variation is smaller than cross-sample variation. This is precisely the "high homogeneity" setting our paper focuses on, which is common in the applications we study (Sec. 5.1 experiment setting).
> 3. **The Denominator and Tightness**: If the condition is violated (low homogeneity), the denominator approaches zero, and the bound goes to infinity (becoming trivial, as PCC $\le 1$). This is expected: when the convex hull is large, the capacity is not restricted by this constraint. The bound is tightest and most relevant when homogeneity is high (small $\tilde{R}$).
>
> ### **Q1: Gradient Comparison Between PCC and MSE**
> We thank the reviewer for this insightful question. To better compare MSE with PCC gradient, we explicitly derive the MSE gradient (as we did for PCC in Sec. 2.3 in revision pdf). The MSE loss has gradient:
> $$
> \frac{\partial L_{\mathrm{MSE}}}{\partial z_{si}} = \frac{2}{S}(\hat y_s - y_s)\alpha_{si}\mathbf{w}^\top(\mathbf{h}_{si}-\mathbf{v}_s).
> $$
>
> and recall the PCC gradient in the paper:
> $$
> \frac{\partial \rho}{\partial z_{si}} = \frac{1}{S\sigma_{\hat y}}\left(\frac{a_s}{\sigma_y} - \rho\frac{b_s}{\sigma_{\hat y}}\right)\alpha_{si}\mathbf w^\top(\mathbf h_{si}-\mathbf v_s)
> $$
>
> Thus, up to a common factor $L_{si}=\alpha_{si}\mathbf w^\top(\mathbf h_{si}-\mathbf v_s)$, the difference between the MSE and PCC gradients is entirely the following global scalar terms:
> $
> g_s^{\mathrm{MSE}} := 2(\hat y_s - y_s),
> \quad
> g_s^{\mathrm{PCC}} := \frac{1}{\sigma_{\hat y}}
> \left(\frac{a_s}{\sigma_y}-\rho\frac{b_s}{\sigma_{\hat y}}\right)\.
> $
>
> Thereby, we clarify that while both effects hinder correlation learning, they contribute differently to the observed plateau.
> 1. **Homogeneity Effect**: The reviewer is correct that data homogeneity suppresses the magnitude of the shared local factor $L_{si}$ (as the aggregated representation becomes less sensitive to attention weights). This indeed weakens the gradients for both MSE and PCC proportionally and makes the optimization generally harder for PCC, but it does not properly explain why PCC plateaus (earlier than MSE). We have clarified this in the revision.
>
> 2. **Optimization Dynamics (PCC-Specific Attenuation)**: This cannot be explained by the homogeneity effect alone and it is unique to the PCC interaction. Theorem 2.2 derives the gradient $\frac{\partial \rho}{\partial z_{si}} \propto \frac{1}{\sigma_{\hat{y}}}$. As MSE optimization succeeds, it drives $\sigma_{\hat{y}}$ to match the target $\sigma_{y}$ (often increasing it from initialization). This creates a paradox: improving MSE (magnitude) dynamically suppresses the PCC gradient (shape), causing the plateau. This occurs regardless of homogeneity.
> Our newly added Corollary 2.1 also rigorously proves that the ratio of the global factors, $\operatorname{RMS}(g_s^{\mathrm{PCC}})/\operatorname{RMS}(g_s^{\mathrm{MSE}})$, decays rapidly at a rate of $\mathcal{O}(1/\sigma_{\hat y}^{3/2})$. As joint training proceeds, MSE optimization drives $\sigma_{\hat y}$ up (Fig. 4). This increase causes the PCC gradient to become negligible relative to the MSE gradient, regardless of the shared homogeneity effects.

---

> ### Author Response · Authors · 2025-11-21
>
> ### **Q2: Effect on High Output Deviation Initialization**
> Thank you for the question and we do observe a plateau with high output deviation.
>
> We clarify the heuristic discussion in Sec. 2 assumes the standard case where the model is typically initialized with small variances (e.g., Xavier/Kaiming) so $\sigma_{\hat{y}}$ starts small and increases under the MSE term, which is what we empirically observe in Fig. 4. The analysis itself, however, does not rely on monotonic increase: the attenuation in Thm. 2.1 is valid whenever $\sigma_{\hat{y}}$ is large.
>
> If we start from an initialization with high prediction deviation, the $1/\sigma_{\hat{y}}$ factor is already small from the beginning, so PCC gradients are weaken earlier. Our analysis in Corollary 2.1 (newly added in revision) shows that the ratio of the PCC gradient magnitude relative to the MSE gradient magnitude decays as $\mathcal{O}(1/\sigma_{\hat{y}}^{3/2})$. If $\sigma_{\hat{y}}$ is large at initialization, this ratio is already small.
>
> Consequently, the optimization would immediately favor MSE reduction (primarily mean-matching) over correlation improvement. This scenario would result in a PCC plateau. While we empirically observe that $\sigma_{\hat{y}}$ typically starts low and increases under standard initialization schemes, this observation illustrates how the model reaches the attenuated state; the attenuation effect itself is governed by the magnitude of $\sigma_{\hat{y}}$ at any point during training.
>
> ### **Q3: Effect of Batch Size on PCC Plateau**
> We clarify that the batch size $S$ does not drive the fundamental conflict causing the PCC plateau (where PCC stalls while MSE improves). The optimization conflict identified in Corollary. 2.1 (newly added in revision version), the decay of the PCC/MSE gradient ratio, is independent of $S$. As shown in Section 2.3, both the PCC and MSE gradients scale equally with $1/S$, so this factor cancels out in the ratio. We empirically validate this in the new **Appendix. H**: synthetic experiments with varying batch sizes ($S \in \{32, 64, 128\}$) consistently exhibit the phenomenon earlier PCC flattening while MSE continues to decrease, confirming the phenomenon is independent of $S$.
>
> ### **Q4: Comment on Read-out Layer Direction**
> Thank you for raising this question. In Theorem. 2.2 the term $\sigma_0/\|\mathbf{w}\|_2$ can be rewritten as
> $
> \frac{\sigma_0}{\|\mathbf{w}\|_2}
> = std_s\left( \left( \frac{\mathbf{w}^\top}{\|\mathbf{w}\|_2} \right)\mu_s \right)
> $
> so the bound depends only on the direction $\mathbf{w}/\|\mathbf{w}\|_2$, not its magnitude.
>
> Geometrically, this measures how much the sample means $\mu_s$ are spread **along the regression direction**.
> If $\mathbf{w}$ aligns with a low-variance direction in representation space, $\sigma_0/\|\mathbf{w}\|_2$ is small and the denominator $\sigma_0/\|\mathbf{w}\|_2 - \widetilde{R}$ is small, yielding a tight PCC-gain ceiling; aligning with a higher-variance direction loosens the bound. The direction of $\mathbf{w}$ acts as a feature selector, guiding the attention mechanism toward the elements most influential for the prediction task.
>
> Since $\mathbf{w}$ is learned from the joint objective, the Theorem. 2.2 says that for whatever prediction direction the model settles on, the achievable PCC gain of **any** convex attention is limited by the geometry of $\{\mu_s\}$ along that direction: which is exactly the homogeneity-driven capacity constraint we highlight.

---

### Author Response · Authors · 2025-12-02
**Summary of Discussion**

We thank all reviewers for their insightful feedback and sincerely appreciate the Area Chair’s thoughtful assessment for the final decision. Below we summarize our contributions and how we addressed the main concerns in the rebuttal.

### **1. Key Contributions:**

**Theoretical Study:** We present the first theoretical analysis of the optimization dynamics and capacity limits of joint MSE+PCC training under attention-based aggregation. We identify a fundamental **optimization conflict** where MSE minimization paradoxically attenuates the PCC gradient (“PCC Plateau”, validated across 8 datasets in App. F) and prove that the achievable correlation for any convex aggregator is **strictly bounded** by within-sample homogeneity.

**Proposed Method:** We propose Extrapolative Correlation Attention (ECA), a backbone-agnostic framework designed to overcome these limitations by restoring gradient magnitude and allowing prediction extrapolation beyond the convex hull.

**Empirical Validation:** Across synthetic, tabular, and challenging high-homogeneity benchmarks (spatial transcriptomics and multimodal sentiment analysis), ECA consistently breaks the PCC plateau and achieves significant correlation gains without degrading MSE compared to standard attention.

### **2. Reviewers Reception:**

We are encouraged by the positive reviews from the reviewers:

**Novelty and Significance:** Reviewers **y2UY**, **4Xnu**, and **pBmc** highlight the novelty and significance of our work, noting that the "PCC plateau" is an interesting, under-theorized phenomenon and that our study is the first to rigorously address this issue of broad interest to the community.

**Principled Theoretical Analysis:** Reviewers **W6BH** and **pBmc** commend our "thorough" and "transparent" theoretical analysis for providing a clean interpretation of the plateau through optimization dynamics and capacity bottlenecks, while reviewer **4Xnu** emphasizes that our solution (ECA) is "principled rather than heuristic" because it is directly motivated by these findings.

**Strong Empirical Validation:** Reviewers **y2UY**, **4Xnu**, and **pBmc** praise the breadth and depth of our experiments. Noting that the consistent improvements across diverse settings, from controlled synthetic datasets to complex spatial transcriptomics, strongly support the "generality and effectiveness" of our approach.


### **3. Clarifications in the Rebuttal and Revision:**
We fully addressed reviewer concerns in the rebuttal, reinforcing the validity and robustness of our main conclusions. We incorporated all clarifications, proofs, and additional experiments into the revised PDF (highlighted in blue). Below, we summarize three major refinements:

**Direct Gradient Comparison:** Following suggestions from reviewers **y2UY**, **pBmc** and **W6BH**, we explicitly analyze the ratio between MSE and PCC gradient (in rebuttal and revision Sec. 2.3). The upper bound $\mathcal{O}(1/\sigma_{\hat{y}}^{3/2})$ better formalizes how MSE-driven growth of $\sigma_{\hat{y}}$ will attenuate the PCC gradient.

**Architectural Depth and Generality:** As discussed with reviewers **y2UY** and **4Xnu**, we clarify that our theory targets the final convex readout (e.g., [CLS] pooling) widely used in standard deep Transformers, regardless of backbone depth. Experiments on deep models (FT-Transformer, ALMT) confirm the plateau persists and is resolved by our ECA (revision Sec. 2.1 and App. E).

**Baseline Coverage:** As suggested by reviewer **4Xnu**, we clarified that our ECA is a modular, plug-in framework complementary to any future non-convex aggregation mechanisms. New experiments show ECA outperforms a negative-weight attention baseline (Cog Attention), and applying our modules (DNPL/DATS) to the Cog Attention baseline further improves its performance.

### **4. Post-rebuttal Scores:**

Reviewer **pBmc** found the clarifications satisfactory and raised the score **4 → 6** (Nov 25);

Reviewer **4Xnu** agreed with the revisions and maintained the score **6**  (Nov 27);

Reviewers **W6BH** and **y2UY** had no further concern following our rebuttal.

---

### Meta-Review · Area_Chair_h7JU · 2026-01-08

**Summary:**

This paper studies the optimisation of attention-based regression models.  It outlines and provides a theoretical analysis of a phenomenon called the PCC plateau, in which the PCC stops improving in training despite the decrease of MSE.

Several of the reviewers commended the work for its novelty and significance.  They also appreciate the work for pointing out the interesting problem and for studying it from both a theoretical and empirical approach.

**Reviewer Concerns:**

(1) simplicity of setup - the analysis is based on a model which has only a single attention aggregation layer with a linear model
(2) rigorousness of theoretical analysis
(3) various clarifications.

This paper improved on (2) and (3) in the rebuttal.  Most of the raised questions on clarifications were addressed.  The only remaining point is that the overall modelling is based on a simple setup of a single attention aggregation layer.  However, the AC is of the belief that this simple model is a reasonable starting point.

**Reviewer Scores:**

pBmc - gave a score of 4 but mentioned they would raise their score to 6.
4Xnu - gave a score of 6.  I don't think this reviewer would raise their score.
W6BH - gave a score of 4.  I am not sure this reviewer would've raised their score, as the authors have answered most of their raised questions in relation to the listed weakness on rigorousness.
y2UY - gave a score of 4.  I think this reviewer would've raised their score, as the rebuttal addresses most of their raised weaknesses and concerns

---

### Decision · Program_Chairs · 2026-01-26

Accept (Poster)